# Single cell transcriptomic landscape of diabetic foot ulcers

Georgios Theocharidis[1,9], Beena E. Thomas[2,9], Debasree Sarkar [2,9], Hope L. Mumme [2], William J. R. Pilcher[2], Bhakti Dwivedi[3], Teresa Sandoval-Schaefer[4], Ruxandra F. Sîrbulescu [5], Antonios Kafanas[6], Ikram Mezghani[1], Peng Wang[1], Antonio Lobao[1], Ioannis S. Vlachos[7], Biraja Dash[8], Henry C. Hsia[8], Valerie Horsley [4], Swati S. Bhasin [2], Aristidis Veves [1,10✉] & Manoj Bhasin[2,10✉]

Diabetic foot ulceration (DFU) is a devastating complication of diabetes whose pathogenesis remains incompletely understood. Here, we profile 174,962 single cells from the foot, forearm, and peripheral blood mononuclear cells using single-cell RNA sequencing. Our analysis shows enrichment of a unique population of fibroblasts overexpressing *MMP1, MMP3, MMP11, HIF1A, CHI3L1,* and *TNFAIP6* and increased M1 macrophage polarization in the DFU patients with healing wounds. Further, analysis of spatially separated samples from the same patient and spatial transcriptomics reveal preferential localization of these healing associated fibroblasts toward the wound bed as compared to the wound edge or unwounded skin. Spatial transcriptomics also validates our findings of higher abundance of M1 macrophages in healers and M2 macrophages in non-healers. Our analysis provides deep insights into the wound healing microenvironment, identifying cell types that could be critical in promoting DFU healing, and may inform novel therapeutic approaches for DFU treatment.

[1] The Rongxiang Xu, MD, Center for Regenerative Therapeutics and Joslin-Beth Israel Deaconess Foot Center, Beth Israel Deaconess Medical Center and Harvard Medical School, Boston, MA, USA. [2] Aflac Cancer and Blood Disorders Center, Children Healthcare of Atlanta, Department of Pediatrics and Biomedical Informatics, Emory University, Atlanta, GA, USA. [3] Winship Cancer Institute, Emory University, Atlanta, GA, USA. [4] Molecular, Cellular and Developmental Biology, Yale University, New Haven, CT, USA. [5] Vaccine and Immunotherapy Center, Department of Medicine, Massachusetts General Hospital, Harvard Medical School, Boston, MA, USA. [6] Lincoln County Hospital, Northern Lincolnshire and Goole NHS Foundation Trust, Scunthorpe, UK. [7] Department of Pathology, Beth Israel Deaconess Medical Center, and Harvard Medical School, Boston, MA, USA. [8] Yale Plastic and Reconstructive Surgery-Wound Center, Yale School of Medicine, New Haven, CT, USA. [9] These authors contributed equally: Georgios Theocharidis, Beena E. Thomas, Debasree Sarkar. [10] These authors jointly supervised this work: Aristidis Veves, Manoj Bhasin. ✉email: aveves@bidmc.harvard.edu; manoj.bhasin@emory.edu

Diabetic foot ulceration (DFU) is a major problem in diabetic patients as more than 15% of them are expected to develop DFUs within their lifetime. DFUs significantly impair quality of life, lead to prolonged hospitalization, and result in more than 70,000 lower extremity amputations per year in the USA alone[1]. Notably, more than half of the patients undergoing amputation due to DFU are expected to die within 5 years, a mortality rate which is higher than most cancers[2]. With the expected increase of Diabetes Mellitus (DM), DFUs will represent an even bigger burden for health systems worldwide and may prove to be one of the costliest diabetes complications[3].

Impaired wound healing leading to the development of chronic wounds in diabetic patients manifests exclusively in the foot in the presence of neuropathy and/or vascular disease[4,5]. Various cell types, including endothelial cells, fibroblasts, keratinocytes, and immune cells play an important role in the wound healing process, but little is understood about their involvement in impaired wound healing in DFU. Dissecting cell differences within the foot ulcers between DFU patients whose ulcers heal and those who fail to heal and go on to develop a chronic ulcer, the differences between DM patients and non-DM healthy controls, and the differences between foot with DFU and intact forearm skin in both DM and healthy subjects, along with differences in blood immune cells, can considerably increase our understanding of DFU pathogenesis/healing.

Single-cell RNA-sequencing (scRNASeq) analysis provides deep insight into cell function and disease pathophysiology by allowing the profiling of the transcriptome landscape of individual cells in heterogeneous tissues. Currently, scRNASeq is widely used in the complex ecosystems of various cancers to map their microenvironment and discover molecular mechanisms and therapeutic targets[6], and concerted efforts of the human cell atlas initiative aim to fully profile all tissues of the human body[7]. Initial studies in our groups have reported that DM and DFU patients have increased inflammatory cells and different fibroblast clusters with a distinctive injury response-associated gene expression profile, which is believed to be the result of DM related chronic inflammation[8]. Spatial transcriptomics (ST) is a more recent method that enables the visualization and quantitation of the transcriptome in individual tissue sections, retaining spatial molecular information unlike scRNASeq[9].

In the present study, we primarily focused on differences between DFU patients whose ulcers heal (DFU-Healers) and those who don't heal (DFU-Non-healers) within 12 weeks. We hypothesize that diabetic patients with impaired wound healing have aberrant gene and protein expression profiles that lead to dysregulation of epithelial remodeling and inflammation pathways. To this end, we investigated the molecular changes via scRNASeq analysis of DFUs and forearm skin biopsies, and peripheral blood mononuclear cells (PBMCs) from patients with healing and non-healing DFUs. As control group we also performed scRNASeq analysis of the foot and forearm biopsies and PBMCs from DM patients with no DFU, and healthy non-DM patients. We also studied different sites of a chronic wound (wound site, wound periphery, and healthy skin) to validate our findings. We finally employed immunostaining and ST on DFU sections, as well as performed in vitro experiments to confirm our most striking findings associated with DFU wound healing.

## Results

**DFU healing is significantly associated with a subset of fibroblasts.** To identify local and systemic factors associated with DFU healing, we examined the cellular landscape of DFUs by scRNASeq analysis of skin specimens from DFU, foot, forearm, and PBMC samples. We analyzed 54 samples from 17 diabetic

patients (11 with and 6 without DFU) and 10 healthy non-DM subjects. The study cohort, objectives, and analysis strategy are outlined in Fig. 1a. In total, we sequenced 174,962 cells (94,325 from foot, 37,182 from the forearm, and 43,455 from PBMC samples) and created a gene expression matrix for each cell, which we used to perform dimensionality reduction by UMAP and graph-based clustering, thereby identifying 37 orthogonal clusters of cells. The expression of established cell-specific marker genes assisted in the annotation of these 37 cell clusters into 21 distinct cell types (Fig. 1b, c). We identified most of canonical cell types observed in the human skin[10,11] and PBMCs[12], namely: smooth muscle cells, SMCs ($TAGLN^+$, $ACTA2^+$); fibroblasts, Fibro ($DCN^+$, $CFD^+$); vascular endothelial cells, VasEndo ($ACKR1^+$); T-lymphocytes, T-lympho ($CD3D^+$); CD14$^+$ monocytes, CD14-Mono (CD14$^+$, $S100A9^+$); differentiated keratinocytes, DiffKera ($KRT1+$, $KRT10^+$); basal keratinocytes, BasalKera ($KRT5^+$, $KRT14^+$); natural killer cells, NK ($CCL5^+$, $GZMB^+$); NK and T cells, NKT ($CD3D^+$, $CCL5^+$); CD16$^+$ monocytes, CD16-Mono ($FCGR3A^+$/$CD16^+$); M1 macrophages, M1-Macro ($IL1B^+$); M2 macrophages, M2-Macro ($CD163^+$); melanocytes and Schwann cells, Melano/Schwann ($MLANA^+$, $CDH19^+$); sweat and sebaceous gland cells, Sweat/Seba ($DCD^+$); lymphatic endothelial cells, LymphEndo ($CCL21^+$), erythrocytes, Erythro ($HBB^+$); dendritic/Langerhans cells, DCs ($GZMB^+$, $IRF8^+$); B-lymphocytes, B-lympho (CD79A$^+$, $MS4A1^+$); plasma cells, Plasma ($MZB1^+$), and mast cells, Mast ($TPSAB1^+$) (Fig. 1c). Comparative analysis of cell type abundance revealed substantial variations in the enrichment across clinical groups (Fig. 1d). Statistical analysis on cellular abundance showed significant variation for HE-Fibro, NKT, plasma and erythrocytes among the clinical groups (Supplementary Fig. 1). The enrichment of mast cells in non-diabetic subjects is in agreement with our previous work demonstrating excessive degranulated mast cells in diabetic skin, which could affect their ability to survive enzymatic digestion and sequencing[13]. Erythrocytes were increased in DFU-Non-healers, most probably due to insufficient RBC lysis during sample processing. Plasma cells were also enriched in DFU-Non-healers, reflecting a possible link of B-lymphocyte differentiation with non-healing wounds. Interestingly, our analysis showed significant heterogeneity in the transcriptome profile of fibroblasts and identified a unique population of fibroblasts that were overrepresented in the samples from DFU-Healers (Fig. 1b, d). We will refer to these as Healing Enriched-Fibroblasts, HE-Fibro, in the rest of the article. Further, the gene signature for each cell cluster was defined by comparing the expression profile of the target cluster with the rest of cells based on non-parametric Wilcoxon Rank Sum test (P-value <0.01 and Fold Change >2) (Fig. 1e). In-depth analysis of the HE-Fibro cell cluster revealed high expression of multiple extracellular matrix (ECM) remodeling (MMP1, MMP3) and immune/inflammation (CHI3L1, TNFAIP6) associated genes (Fig. 1f). MMP1 (Matrix Metalloproteinase-1) interacts with CD49b[14,15], an integrin alpha subunit involved in cell adhesion and cell-surface-mediated signaling in T, NK, and NKT cells[16], fibroblasts, and platelets. CHI3L1 (Chitinase-3-Like Protein 1) is a secreted glycoprotein that has been previously associated with pathogenic processes related to inflammation and ECM remodeling[17]. TNFAIP6 (tumor necrosis factor alpha induced protein 6) is known to be involved in ECM stability and cell migration, and its expression is correlated with proteoglycan synthesis and aggregation[18]. This protein has shown anti-inflammatory effects in various models of inflammation, which suggest that it is a component of a negative feedback loop capable of downregulating the inflammatory response[19]. The distinct and previously undescribed subtype or state of fibroblasts, HE-Fibro, with overexpression of matrix remodeling, immune and inflammatory genes, may contribute to successful wound

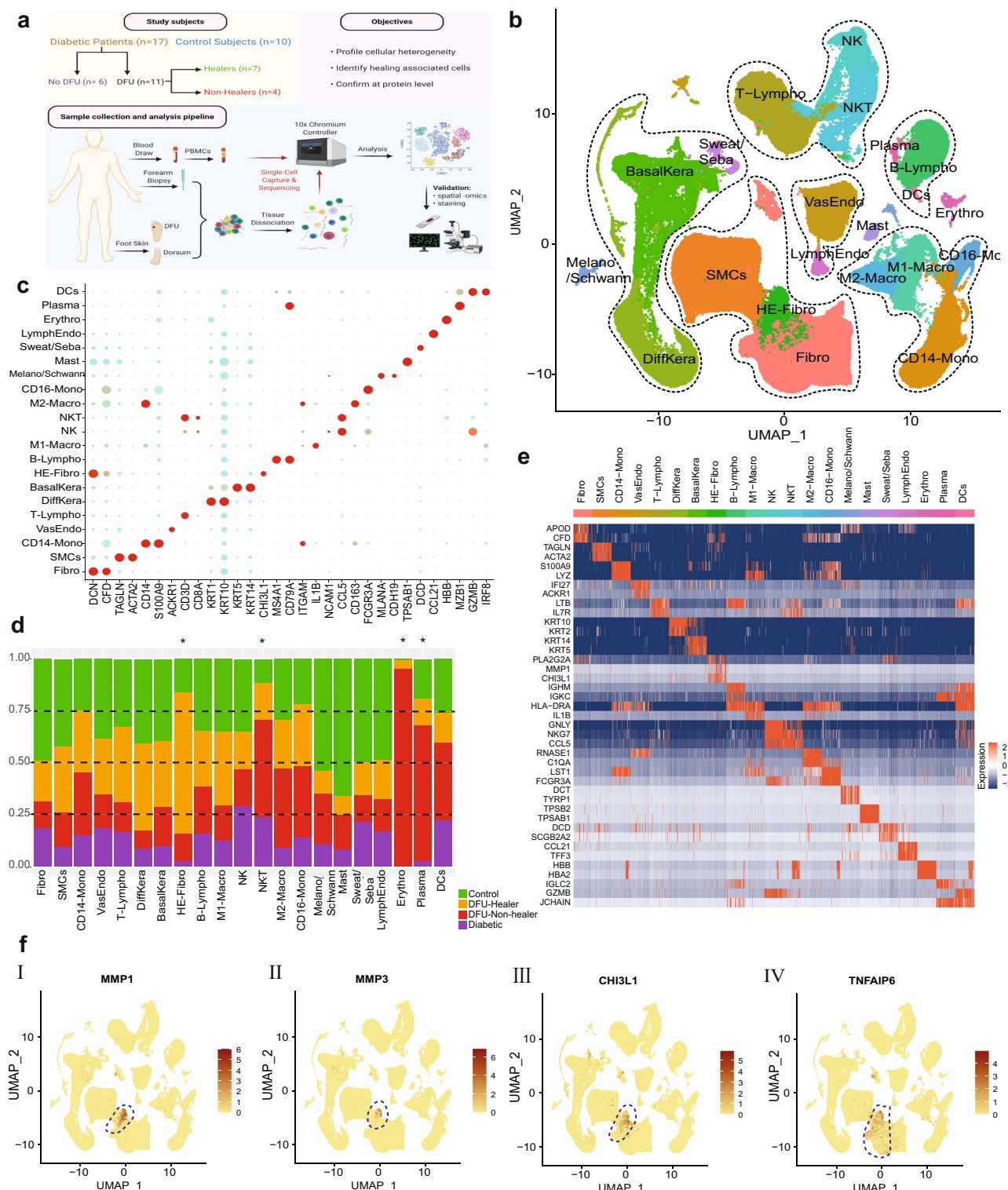

repair in DFU-Healers. The top 10 overexpressed genes from the annotated cell clusters are included in Supplementary Dataset 1.

## Exploring cellular heterogeneity across different anatomical sites.

To assess tissue specific cellular heterogeneity along with gene expression and molecular pathway alterations, we generated the UMAP rendering split based on foot, forearm, or PBMC samples (Fig. 2a). The analysis depicted significant variations in the abundance of cell types based on anatomical sites (Fig. 2b).

Fibroblasts, smooth muscle cells, melanocytes, sweat gland cells, vascular and lymphatic endothelial cells were enriched in the foot samples. The analysis on keratinocytes revealed a predominance of basal and differentiated keratinocytes in the foot and forearm samples, respectively (Fig. 2a, b). Interestingly, $99.94 \pm 1.58\%$ (mean ± stderr) of HE-Fibro cells, were identified in the foot samples (Fig. 2a, b, Supplementary Table 1), indicating that these are foot specific cells. The analysis demonstrated that a significant fraction of immune cells was contributed by the PBMC samples. $CD14^+$ ($98.47 \pm 1.06\%$) and $CD16^+$ ($99.63 \pm 1.11\%$) monocytes

**Fig. 1 Single-cell RNA sequencing mediated identification and characterization of unique healing enriched fibroblasts in diabetic foot ulcers (DFUs).** **a** Schematic overview of the study design and number of samples per clinical group. **b** Uniform Manifold Approximation and Projection (UMAP) embedding of the entire dataset consisting of 174,962 cells. The cells are colored by orthogonally generated clusters, and labeled by manual cell type annotation (HE-Fibro: healing enriched fibroblasts, Fibro: fibroblasts, SMCs: smooth muscle cells, BasalKera: basal keratinocytes, DiffKera: differentiated keratinocytes, Sweat/Seba: sweat and sebaceous gland cells; Melano/Schwann: melanocytes and Schwann cells; Mast: mast cells; VasEndo: vascular endothelial cells; LymphEndo: lymphatic endothelial cells; CD14-Mono: CD14+ monocytes, CD16-Mono: CD16+ monocytes, M1-Macro: M1 macrophages, M2-Macro: M2 macrophages, Erythro: erythrocytes, NK: natural killer cells, T-Lympho: T-lymphocytes, NKT: NK cells and T lymphocytes; B-Lympho: B-lymphocytes, Plasma: plasma cells, DCs: dendritic cells). Dotted lines are drawn around cell groups of similar lineages. **c** Dot plot showing expression of different cell type-specific marker genes, used to annotate the cell types. Size of dots indicates percentage of cells in each cell cluster expressing the marker gene; color represents averaged scaled expression levels; cyan: low, red: high. **d** Stacked bar plots showing the proportion of different cell types across the four clinical groups. Green: Healthy subjects, orange: DFU-Healers, red: DFU-Non-healers, purple: Diabetic patients. Cell types with significant differences among the clinical groups are marked with an asterisk. The bar plots for individual cell types are presented in Supplementary Fig. 1. **e** Heatmap showing the top highly expressed (red) genes in each of the cell clusters. **f** Feature plots depicting the expression of key genes (I) *MMP1*, (II) *MMP3*, (III) *CHI3L1*, (IV) *TNFAIP6*, that were significantly overexpressed in the healing enriched fibroblasts associated with healing of DFUs. The schematic on (**a**) was created with BioRender (BioRender.com).

were observed predominantly in the PBMC samples. Dendritic, NK and NKT cell populations were also predominantly present in PBMC samples. The highest abundance of M2 macrophages (71.28 ± 1.27%) and mast cells (82.51 ± 1.49%) was observed in the foot samples, with lower proportions in the forearm samples, and the lowest in the PBMCs. On the adaptive immune system side, 31.27 ± 0.27% of T-lymphocytes came from the foot samples, 12.42 ± 0.35% from the forearm, and the remaining 56.31 ± 1.21% from the PBMCs (Fig. 2b, Supplementary Table 1). In contrast, more than half of plasma cells were derived from foot (66.01 ± 1.89%), and the remaining were almost equally proportioned between forearm and PBMCs. Most of B-lymphocytes (84.49 ± 1.63%) originated from PBMCs.

To more closely examine the gene expression landscape of cells with differential abundances between foot and forearm i.e., fibroblasts and keratinocytes, we performed comparative analysis on their transcriptome profiles (Fig. 2c–f). The foot fibroblasts exhibited upregulation of multiple genes associated with ECM remodeling and immune response. This may be attributed to the enrichment of HE-Fibro population in the foot samples (Fig. 2c). Genes that were overexpressed in foot fibroblasts include the gene for Wnt signaling antagonist, secreted frizzled-related protein 4 (*SFRP4*) and genes directly related to ECM organization, asporin (*ASPN*) and tenascin C (*TNC*). Wnt signaling is crucial for effective wound healing[20,21] and its modulation is closely linked with TGFβ expression[22], which is in line with the enhanced expression of *TGFB1* in these fibroblasts. Tenascin C is known to upregulate TGFB1 as well as promote expression of type I collagen in fibroblasts, which is essential for maintaining ECM integrity[22]. The cellular development (*TSPAN8*, *WIF1*) and immune cell trafficking (*CCL19*) related genes were significantly overexpressed in the fibroblasts from the forearm (Fig. 2c). Pathway analysis on the foot fibroblasts' differentially expressed genes (DEGs) revealed significant (*P* value <0.01) activation of ILK, leukocyte extravasation signaling, RhoA signaling, and actin cytoskeleton signaling (Fig. 2d). The comparative analysis between foot and forearm keratinocytes showed significant upregulation of basal (*KRT6A*, *KRT16*, *KRT17*) and differentiated (*KRT2*, *KRT10*) keratinocyte associated genes in the foot and forearm samples, respectively (Fig. 2e). This discrepancy can be explained by the fact that forearm biopsies represent unwounded tissues with fully stratified epidermis, as opposed to foot samples that include DFUs with partially formed epithelium, and therefore fewer differentiated keratinocytes. Moreover, the differences between plantar glabrous skin and forearm hairy skin could contribute to the disparity[23]. In addition to upregulation of alarmins like *KRT6A/16/17* in the foot samples[24], we also observed upregulation of inflammatory molecules including

*S100A8* and *S100A9*, known to activate the immune system in response to skin injury[25]. Further pathway analysis on foot keratinocyte DEGs uncovered significant activation of immune and inflammatory pathways including ILK and IL-8 signaling (Fig. 2f).

**Systemic dysregulations revealed by comparative analyses of PBMCs across clinical groups.** To better understand the impact of DFU at a systemic level, we performed separate analysis on PBMC samples alone from the four clinical groups, viz Healthy (healthy subjects without DM), DFU-Healer (DM patients with healing DFUs), DFU-Non-healer (DM patients with non-healing DFUs), and Diabetic (DM patients without DFU) (Fig. 3a). The cell annotation was done using well-established marker genes (Supplementary Fig. 2). The DFU-Healers were observed to have higher proportions of naive and early differentiated progenitor T-lymphocytes, T-lympho, expressing *CCR7*, shown to have a role in activation of various T-cell subsets[26] (Fig. 3b). On the other hand, DFU-Non-healers had a higher proportion of cytotoxic NKT cells (*IL7R*, *GZMB*, *KLRD1*), indicating a shift in T-cell subpopulations correlating with DFU healing (Supplementary Fig. 3). We observed statistically significant higher *CCR7*+ T-lympho cells to NKT cells ratio (*P* value < 0.01) in the DFU-Healers as compared to DFU-Non-healers and DM patients without DFU, indicating the association of these T-cells with successful wound healing (Fig. 3b). A significantly higher proportion of *CCR7*+ *CD8*+ T cells (cluster CD8T2 in Fig. 3a) was also observed in DFU-Healers as compared to DFU-Non-healers (Fig. 3c). Further DEGs analysis on these T-lympho, CD8T2 and NKT cells indicated overexpression of T-cell-specific genes like *IL7R*, *TCF7*, and *CCR7* in the DFU-Healers, whereas DFU-Non-healers overexpressed NKT lineage genes like *NKG7*, *GNLY*, *CCL5*, and *KLRD1* (Fig. 3c). Pathway analysis on these T/NKT cells DEGs demonstrated inhibition of key immune and inflammation pathways including IL-6, IL-8, CD28 Signaling in T helper cells, and iCOS-iCOSL pathways and activation of RhoGDI and EIF2 signaling in the DFU-Healers, as compared to DFU-Non-healers at the systemic level (Fig. 3d). Further systems biology analysis revealed inhibition of several upstream regulators of immune pathways including CD44, TGFB1, CCL5, and NFKBIA in the T-cells from PBMCs of patients with healing DFUs (Fig. 3e). This was in accordance with the observed reduced gene expression of *NFKBIA*, *CCL5*, and *TGFB1* in DFU-Healers compared to high expression in DFU-Non-healers (Fig. 3f). In aggregate, these results underscore the enrichment of naive T-cells with a prevalence of immune inhibitory pathways and processes for DFU-Healers, and a state of chronic inflammation for DFU-Non-healers, at the systemic level.

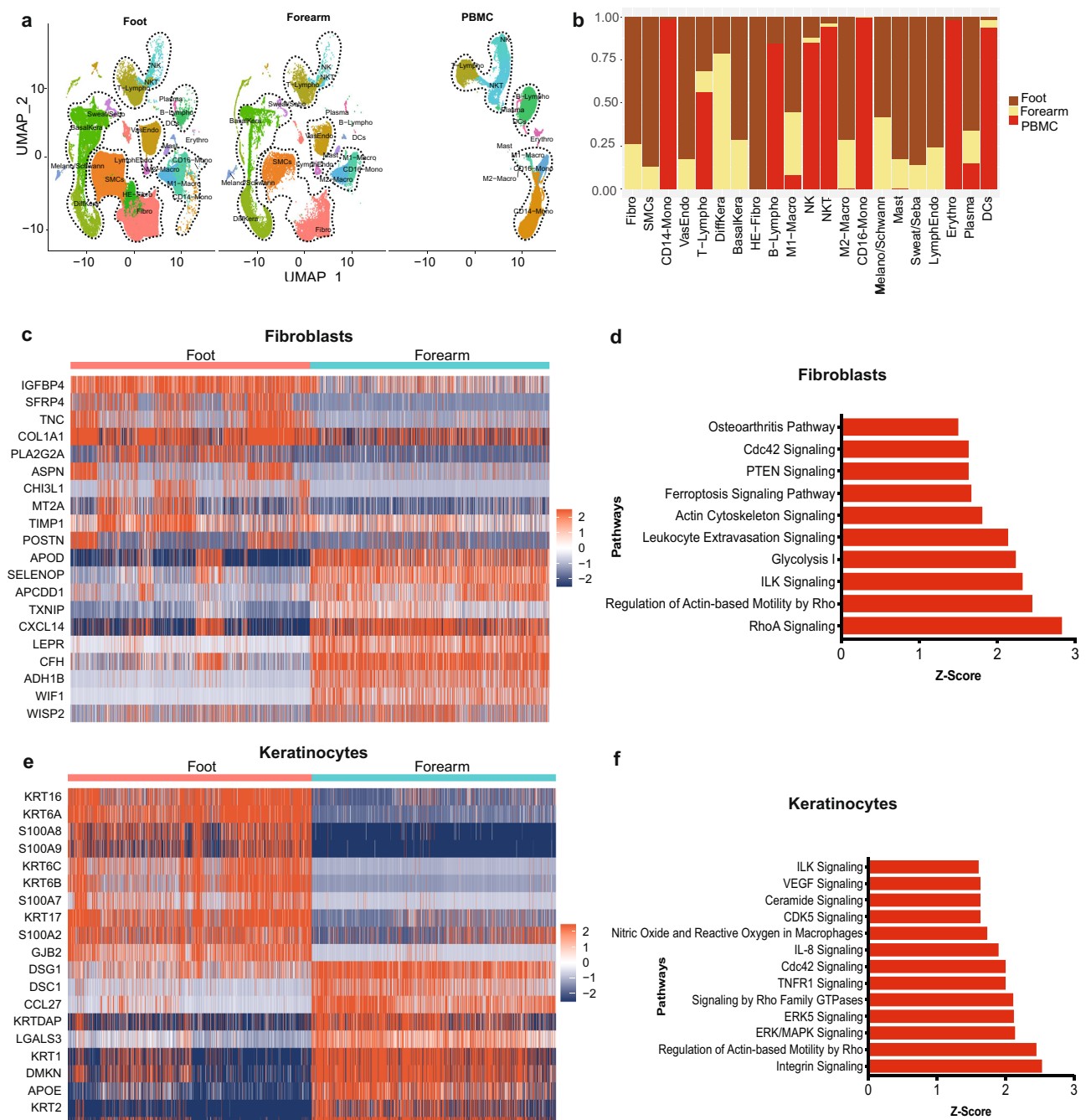

**Fig. 2 Comparative single-cell transcriptome analysis profiles of foot, forearm, and PBMCs, delineating gene signatures, and biological pathways across anatomical sites. a** Split UMAP of Foot, Forearm, and PBMC samples. The cell clusters were annotated manually according to various canonical and novel cell types based on expression of specific markers (as described in Fig. 1b, c). Dotted lines are drawn around cell groups of similar lineages. **b** Stacked bar plots showing the proportion (y-axis) of different cell type from foot, forearm, and PBMC. Dark brown: foot, beige: forearm, red: PBMCs. **c** Heatmap showing significantly differentially expressed genes between foot and forearm fibroblast cell clusters. Relative gene expression is shown in pseudo color, where blue represents downregulation, and red represents upregulation. **d** Pathway enrichment analysis on genes that are significantly differentially expressed between foot and forearm cell fibroblast clusters. The pathways analysis was performed using Ingenuity Pathways analysis (IPA) tool that calculate significance of impact on pathways using one-tailed Fisher's exact test and Z-score. The pathways with *P* value < 0.01 and Z score >2 were considered significantly activated. **e** Heatmap showing significantly differentially expressed genes in keratinocytes cell clusters between foot and forearm samples. **f** Pathway enrichment analysis on genes that are significantly differentially expressed between foot and forearm keratinocytes cell clusters.

**T, NK, and NKT cells exhibit distinct cell subpopulations in DFU-Healers and DFU-Non-healers**. The focused sub-clustering analysis on the T, NK, and NKT cell populations identified 17 sub-clusters (Supplementary Fig. 4a). *CD4+* (sub-clusters 0, 4, 10) and *CD8+* (sub-cluster 14) naive T-cells (*CCR7+, LEF1+*), that can

self-renew and proliferate readily into other T-cells, were enriched in DFU-Healers (Supplementary Figs. 4b, c and 5c, d). Cluster 6, *CD8+* effector T cells (*CCL5+, GZMB+, IL32+, GZMK+*), enriched in DFU-Healers, also expressed higher levels of *CD27*, a key molecule in generation and maintenance of T-cell immunity[27].

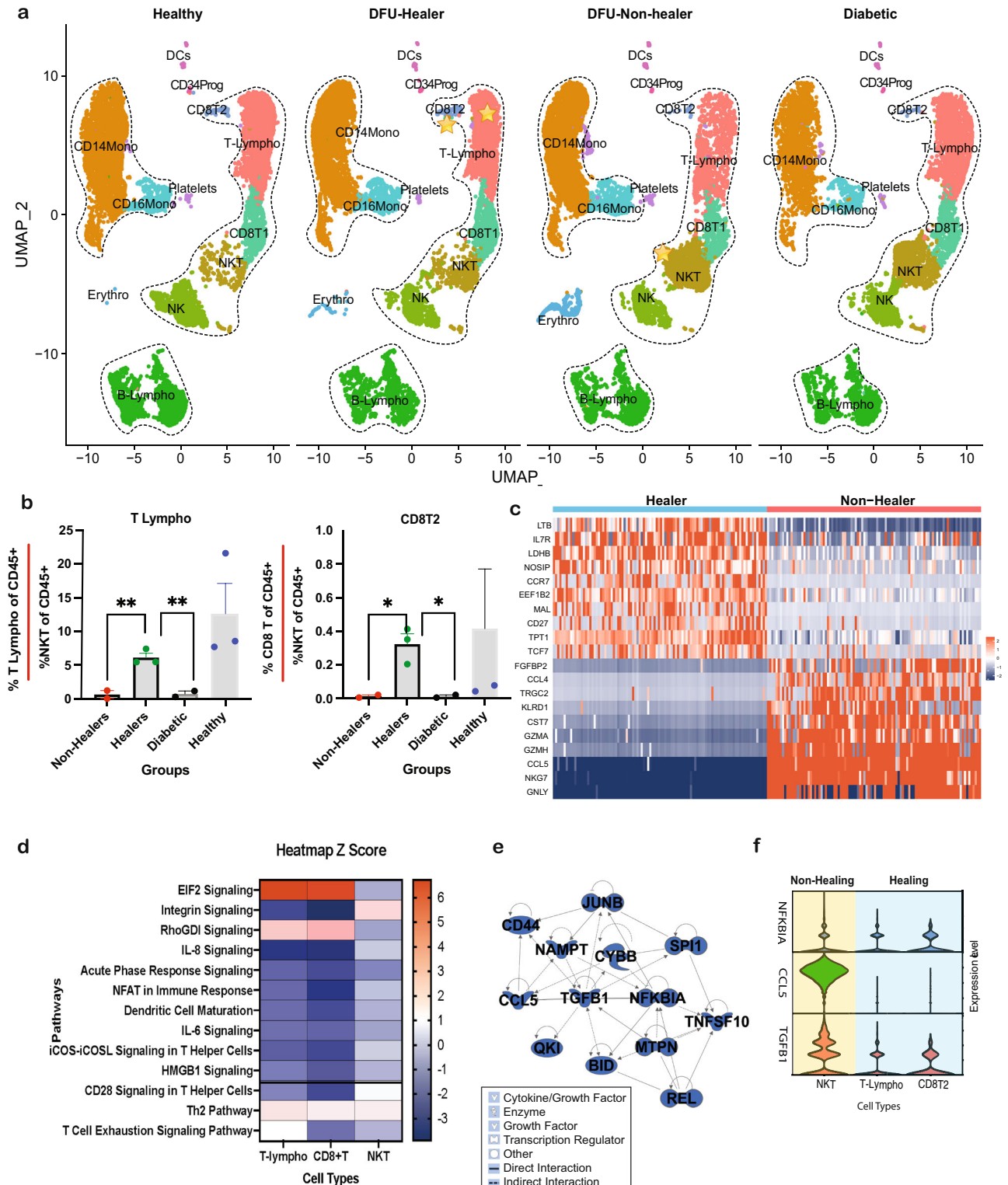

NKT (*CD8+, CCL5+, GZMB+, IL32+, GZMH+*) cells sub-clusters 5 and 7 were enriched in Diabetic and DFU-Non-healer groups, respectively (Supplementary Figs. 4a, b and 5c, d). The DFU-Non-healer enriched sub-cluster 7 also had high expression of T-cell exhaustion marker, *TIGIT* (Supplementary Fig. 5d). Sample site-specific split t-SNE plots revealed separate clustering of T, NKT and NK cells from skin and PBMCs (Supplementary Fig. 5a). Sub-clusters 1, 2, 9, 11, and 13, expressing activation markers *CD69* and *CD44* were largely made up of skin samples

derived from foot (Supplementary Fig. 5a–d). Clusters 1 and 9 were positive for T-cell exhaustion markers (*TIGIT+, HAVCR2+, LAG3+*) (Supplementary Fig. 5c). In PBMCs, DFU-Healers appeared to have more non-polarized central memory and naive T-cells (Supplementary Fig. 4b, c; Supplementary Fig. 5). *CD27* which characterizes central memory T-cells that lack immediate cytotoxicity[28], was also more in the DFU-Healers (Supplementary Fig. 5c). In contrast, DFU-Non-healers were enriched with cytotoxic NKT cells (cluster 7), expressing *GZMH, GZMA,* and

**Fig. 3 Comparative transcriptome profiles analysis of PBMCs in different clinical groups, uncovering differences in systemic immune landscape associated with wound healing response in DFUs. a** UMAP dimensionality reduction embedding of PBMCs from DFU-Healers, DFU-Non-healers, Healthy subjects, and non-DFU DM patients. The identified cell types were DCs: dendritic cells; VasEndo: vascular endothelial cells; T-lympho: T lymphocytes; CD8T1: CD8$^+$ T lymphocytes cluster 1; CD8T2: CD8$^+$ T lymphocytes cluster 2; NK: natural killer cells; NKT: natural killer and T cells; B-lympho: B lymphocytes; CD14Mono: CD14$^+$ monocytes; CD16Mono: CD16$^+$ monocytes. Dotted lines are drawn around cell groups of similar lineages. **b** Bar plots showing percentage of T-lymphocytes (T-lympho) and CD8$^+$ T lymphocytes cluster 2 (CD8T2) per percentage of NKT cells in the CD45$^+$ subset of cells across various clinical groups. DFU-healers depict significantly higher ratio of T-lympho and CDT2 cell cluster in comparison to DFU-Non-healers and Diabetic. Data represent the mean and standard error of mean (SEM) values from $n = 2$ Non-Healers, $n = 3$ Healers, $n = 2$ Diabetic and $n = 3$ Healthy subjects. $p = 0.01$ for Healers vs Non-Healers and $p = 0.006$ for Healers vs Diabetic in T-Lympho; $p = 0.036$ for Healers vs Non-Healers and $p = 0.035$ for Healers vs Diabetic in CD8T2 using two-sided Welch's $t$-test. **c** Heatmap showing significant DEGs in Healers compared to Non-healers in the T-lympho, CD8T2 and NKT cell clusters. **d** Biological pathways that are significantly ($P$ value < 0.01) activated (Z score >1.5) /inhibited (Z score < −1.5) in T-lympho, CD8T2 cells of Healers in contrast to NKT cells of Non-healers. Activation and inhibition of key upstream regulators is shown in pseudo color, where blue represents inhibition, and red represents activation. **e** Upstream regulatory molecules significantly inhibited (blue) in the T-lympho and CD8T2 cells of Healers as compared to Non-healers at the systemic level. Legend shows shapes and lines annotation for the regulatory network. **f** Violin plots showing expression levels of 3 key upstream regulator molecules *NFKBIA, CCL5,* and *TGFB1,* in the NKT, T-Lympho, and CD8T2 clusters.

*GZMB* (Supplementary Figs. 4b and 5c). These granzyme molecules have been previously implicated in impaired wound healing development by promoting chronic inflammation, vascular dysfunction, and reduced cell adhesion[29]. We also observed a unique CD4$^+$ cluster (cluster 10, Supplementary Fig. 4a) that was predominantly present in DFU-Non-healers and enriched for *GIMAP1* and *GIMAP4,* both shown to be implicated in T helper cell differentiation towards the Th1 lineage[30]. In the Diabetic group, a *GZMH$^+$, GNLY$^+$,* and *CCL5$^+$* expressing cluster (cluster 5) was prominent, pointing toward the presence of specialized DM associated NKT cells. CCL5/RANTES, a potent chemoattractant of immune cells, has been reported to be strongly downregulated in DFUs compared to acute wounds, and could represent a potential therapeutic target[31]. In summary, while skin samples derived T/NKT cells did not show significant differences between DFU-Healers and DFU-Non-healers, potentially due to the low number of recovered T-cells, a definitive enhancement of naive T-cells was seen in PBMCs of DFU-Healers compared to more cytotoxic NKT cells in DFU-Non-healers.

**Analysis of foot ulcer cells reveals the significance of localized inflammation in diabetic wound healing.** To map the transcriptome and cellular landscape of the site for DFUs, we performed focused analysis on single-cell profile of 94,325 cells from 26 foot samples. Split UMAP analysis indicated differential abundance of cell types among the four clinical groups (Fig. 4a, Supplementary Fig. 6). The DFU-Healers had a significantly higher number of HE-Fibro cells ($P$ value <0.05) as compared to DFU-Non-healers, Diabetic patients and non-DM healthy controls (Fig. 4b). Additionally, the DFU-Healer group also showed a significantly higher proportion of M1 macrophages (classically activated macrophages that promote inflammation) than M2 macrophages (alternatively activated macrophages with anti-inflammatory properties), as opposed to DFU-Non-healers (Fig. 4c). Also, a group of SMCs, SMC2, with overexpression of proliferation markers *CENPF, PTTG1, MKI67,* and *TOP2A* was significantly enriched in DFU-Healers (Fig. 4d, Supplementary Dataset 2), highlighting the presence of a highly proliferative SMC cluster in healing DFUs. Other cell types also exhibited variation across clinical groups but did not achieve statistical significance due to intragroup variation among patients (Fig. 4e). DEGs analysis on DFU-Healers vs. Non-healers and M1 macrophages vs. M2 macrophages identified a signature comprising of 195 genes that were differentially expressed in M1 macrophages from DFU-Healers (Fig. 4f). DFU-Healer enriched macrophages overexpressed inflammatory genes including *IL1B, S100A8,* and *S100A9* to mount an acute inflammatory

response for promoting wound healing. On the other hand, DFU-Non-healers macrophages overexpressed genes from the complement system like *C1QA/B/C,* which are associated with M2 macrophage-like anti-inflammatory responses[32] (Fig. 4f). Immunofluorescent staining of healing and non-healing DFUs with pan-macrophage marker CD68, M1 markers IL1B and S100A8 and M2 markers DAB2 and CD163 confirmed more M1 associated macrophages in DFU-Healers with both M1 markers showing increased presence and more M2 macrophages in DFU-Non-healers (Supplementary Fig. 7a–f).

Pathway analysis showed activation of the IL17 signaling pathway, a known regulator of inflammatory response[33], in DFU-Healers (Fig. 4g). The upstream regulators activated in DFU-Healers included HIF1A, TNF, STAT5a/b, TLR7, TLR9, and IL17R/C (Fig. 4h), whereas SOX4, TGFB1, and NANOG, were inhibited (Fig. 4i). Immunofluorescent staining with IL17 and HIF1A antibodies showed a trend for more IL17$^+$ cells ($p = 0.06$) and higher HIF1A expression ($p = 0.09$) in DFU-Healers (Supplementary Fig. 7g–i).

Similar analyses were also conducted on the forearm cells (Supplementary Figs. 8a and 9). Differentiated keratinocytes were enriched in DFU-Healers compared to DFU-Non-healers (Supplementary Fig. 8b). We found that *LGALS7* or Galectin-7, which has been previously implicated in keratinocyte migration during re-epithelialization of wounded epidermis[34], was the top differentially expressed gene in the forearm keratinocytes of DFU-Healers (Supplementary Fig. 8c).

**Healing associated fibroblasts drive DFU healing by promoting matrix remodeling and inflammatory response.** To further delineate the role of fibroblasts in wound healing, we performed focused analysis on fibroblasts that produced 14 sub-clusters, representing different molecular states or subtypes of fibroblasts (Fig. 5a, Supplementary Dataset 3 for top 10 marker genes). The majority of sub-clusters showed distinct expression profiles indicating heterogeneity in the fibroblast population (Fig. 5b). Sub-clusters 0, 1, 2, and 5 comprised most of the cells from unwounded skin. Sub-cluster 0 was characterized by the expression of reticular fibroblast marker *MGP*[35] and multiple adipocyte-associated genes (*APOE, APOD, CFD*), consistent with the enhanced adipogenic potential of these cells[36]. Sub-clusters 2 and 5 contained cells expressing papillary fibroblast markers *PTGDS, APCDD1,* and *COL23A1*[37,38], while sub-cluster 1 was enriched for *WISP2, PI16, SLPI,* and *SFRP2,* which describe fibroblasts residing both in the papillary and reticular dermis, and are believed to contribute to ECM homeostasis[39–41]. The evaluation of cellular makeup of clusters unveiled a higher proportion of cells (58–90%) from DFU-Healers in specific sub-clusters; clusters

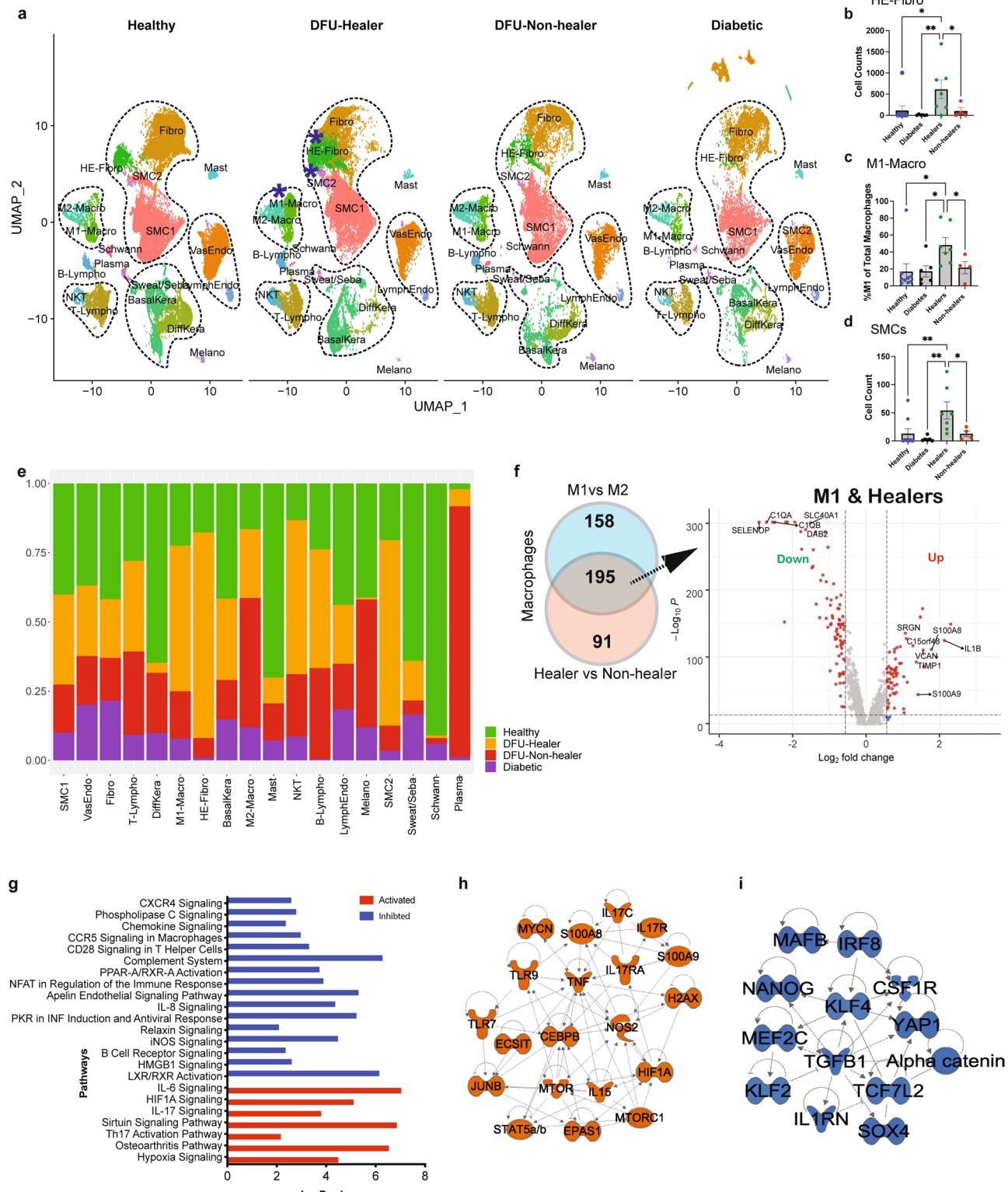

3, 4, 6, and 13. These four sub-clusters represent four heterogeneous states or subtypes of the HE-Fibro (Fig. 5a, marked area). Further generation of gene signatures for these sub-clusters revealed that cluster 3 was significantly enriched with cells expressing genes related to ECM remodeling including *MMP1* and *MMP3* (Fig. 5b). Matrix metalloproteinases MMP1 and MMP3 have been well-known early responders to tissue injury, actively regulating the inflammatory phase of healing by

degradation of the ECM, stimulating leukocyte infiltration for resolution of inflammation and transition to the proliferative phase[42]. Cluster 4 exhibited overexpression of *POSTN* and *ASPN* (Fig. 5b) that are associated with ECM signaling, adhesion and migration. POSTN (Periostin) is a ligand for alpha-V/beta-3 and alpha-V/beta-5 integrins and supports adhesion and migration of epithelial cells[43], and has been shown to play a regulatory role in fibroblast proliferation and inflammation[44,45]. ASPN (Asporin) is

**Fig. 4 Comparative analysis of transcriptome profiles of foot samples in the different clinical groups, elucidating differences in cell type composition, gene expression, and biological pathways. a** UMAP dimensionality reduction embedding of foot cells from DFU-Healers, DFU-Non-healers, Healthy subjects, and non-DFU DM patients. The cellular clusters depicting significant enrichment in the healers are marked with blue asterisks. Dotted lines mark cell groups of similar lineages. Comparative analysis depicted **b** HE-Fibro, **c** M1 macrophages, and **d** SMC2 cellular enrichment in the foot sample from DFU healers. Data represent the mean and SEM values from $n = 9$ Healthy, $n = 6$ Diabetic, $n = 7$ Healer, and $n = 4$ Non-healer subjects. Two-sided Welch's $t$-test was used; $p = 0.013$ for Healthy vs Healers, $p = 0.007$ for Diabetes vs Healers and $p = 0.006$ for Healers vs Non-healers in (**b**); $p = 0.026$ for Healthy vs Healers, $p = 0.017$ for Diabetes vs Healers and $p = 0.042$ for Healers vs Non-healers in (**c**); $p = 0.005$ for Healthy vs Healers, $p = 0.002$ for Diabetes vs Healers and $p = 0.02$ for Healers vs Non-healers in (**d**). **e** Stacked bar plots showing the proportions of different cell types across the different clinical groups (green: Healthy subjects, orange: DFU-Healers, red: DFU-Non-healers, purple: non-DFU DM patients). **f** Venn diagram analysis to compare genes that are differentially expressed between M1 and M2 macrophages and between Healers vs. Non-healers. The comparison identified 195 genes that are differentially expressed in M1 macrophages from DFU-Healers. Volcano plot showing the genes that are significantly differentially expressed (red dots) in M1 macrophages of Healers (Benjamini–Hochberg corrected $P$-value <0.00001, FC > 1). **g** Selected biological pathways that are significantly ($P$ value <0.01) affected in the healing associated M1 macrophages. Each bar represents a pathway with significance of enrichment determined using the one-tail Fisher's exact $t$-test ($-\log10 P$ value is shown on primary X-axis). The directionality of each pathway is depicted using a pseudo color (red for activated, blue for inhibited). Regulators that are significantly activated (**h**) and inhibited (**i**) in the M1 macrophages from Healers. The activation and inhibition of pathways was measured based on Z-score calculation using the IPA platform.

an ECM protein that has been found to inhibit TLR2- and TLR4-induced NF-κB activity and pro-inflammatory cytokine expression in macrophages[46]. TLR4 mediated inflammation drives the synergistic effect of hypoxia and hyperglycemia on impairment of diabetic wound healing[47], hence overexpression of *ASPN* might be an important determining factor for healing of DFUs. In a recent study, a distinct *ASPN* and *POSTN* enriched cluster of fibroblasts was described as mesenchymal and shown to have a more reticular dermis localization[40]. These sub-clusters (3, 4, 6, and 13) were also enriched with genes like *IL6, CHI3L1, PLA2G2A*, and *TIMP1*, commonly associated with an inflammatory signature (Fig. 5b). Based on our analysis, we identified a healing associated fibroblast signature consisting of ECM remodeling and inflammatory response-related genes: *MMP1, MMP3, IL6, CHI3L1, ASPN, POSTN*, and *PLA2G2A*.

Further analysis of these fibroblasts revealed that *IL6/TIMP1/PLA2G2A* and *CHI3L1* transcripts were detected simultaneously in ~38% of the cells suggestive of a common regulatory mechanism in HE-Fibro. We also noticed that 99.8% of *CHI3L1* expressing cells exhibited significant expression of at least one of the ECM remodeling genes including *MMP1, MMP3, MMP11*, indicating a role of these genes in tissue repair. Based on these preliminary results we posit that CHIL3L1 is one of the key players in driving the healing phenotype of HE-Fibro by expressing pro-inflammatory and ECM genes together to improve wound repair. Several lines of evidence have previously implicated *CHI3L1* in dampening of chronic inflammation[48], promoting M1 macrophage activation[49] and stimulating fibroblast proliferation[50] and ECM remodeling[51]. DEGs analysis on these DFU-Healers vs. other fibroblast clusters revealed some ubiquitous markers (*HIF1A, TNFAIP6*) that are overexpressed in HE-Fibro cells (Fig. 5).

Pathway analysis indicated the activation of multiple immune and inflammatory pathways including IL6, HIF1A, and ILK signaling in the fibroblasts from DFU-Healers (Fig. 5c). Moreover, upstream regulator genes like TNF, HIF1A, and IL6, were activated (Fig. 5d) in the DFU-Healers. HIF1A (Hypoxia-inducible factor 1-alpha) is a master-regulator that activates multiple factors to enhance wound healing by promoting cellular motility and proliferation, angiogenesis, re-epithelialization, and cell survival[52]. HIF1A also upregulates IL6 expression by binding to its promoter region[53], thereby promoting inflammation and cell proliferation.

IL6 is a pleiotropic cytokine that plays a vital role in wound healing, as studies have shown delayed and impaired wound closure in IL6 knockout mice[54–56]. It is noteworthy that in our previous work with a large animal model of diabetic wound

healing[57], IL6 levels post-injury were attenuated compared to acute non-diabetic wounds, suggesting that increased IL6 is advantageous for DFUs. Another notable gene in the molecular interaction network is TNF (Fig. 5d), a potent pro-inflammatory cytokine that has been previously implicated in the wound healing process[58], and is known to be elevated shortly after wounding. TNF has also been shown to upregulate the expression of MMP1 and MMP3 in human dermal fibroblasts via NFκB/p65 activation[59]. To corroborate the activation of central regulators TNFA and IL6 at the protein level, we probed healing and non-healing DFUs and measured the percentage of area stained to find significantly higher IL6 expression ($p = 0.001$) and a trend for higher TNFA expression ($p = 0.08$) (Supplementary Fig. 7j–l) in DFU-Healers.

**Deciphering communication among Healing associated fibroblasts.** Further, to determine possible communication among heterogeneous healing associated fibroblasts (HE-Fibro sub-clusters 3, 4, 6, and 13; Fig. 5a), we performed ligand, receptor, and target gene co-expression analysis using the NicheNetR algorithm[60]. NicheNet predicts which ligands from one or more cell population(s), termed "sender/niche", will most likely affect gene expression in interacting cell population(s), termed as "receiver/target". Also, this algorithm can predict which specific target genes in the "receiver" cell populations are affected by the predicted ligands in the "sender" cell population(s). As sub-cluster 3 of the HE-Fibro was enriched in both inflammatory and ECM remodeling genes (*IL6, CHI3L1, MMP1*), critical to the healing process, it was selected as "sender" cell population to generate ligand candidates, while the remaining HE-Fibro sub-clusters (4, 6, and 13) were treated as receiver cells (Supplementary Fig. 10). To filter out non-specific ligand and receptors, we also included control fibroblast sub-clusters (0, 2, and 5), enriched in healthy non-DM and diabetic without DFU patients, as receiver cells. The analysis identified multiple ligands including *IL6, CCL2*, and *TIMP1* with high correlation between differential expression of ligands in "sender" fibroblasts and their target genes in the "healer" fibroblasts (sub-clusters 4, 6, and 13) but not in the "control" fibroblasts (sub-clusters 0, 2, and 5) (Fig. 5e) This indicates that fibroblasts from sub-cluster 3 are primarily interacting with the other three sub-clusters of HE-Fibro subset enriched in DFU-Healers rather than the "control" fibroblasts subset enriched in healthy non-DM and diabetic with no-DFU patients. *FN1* was enriched in all the HE-Fibro sub-clusters (3, 4, 6, and 13), while *IL6, MMP13, CCL2, PTGS2*, and *VEGFA* were enriched in only HE-Fibro sub-cluster 3 (Supplementary Fig. 11).

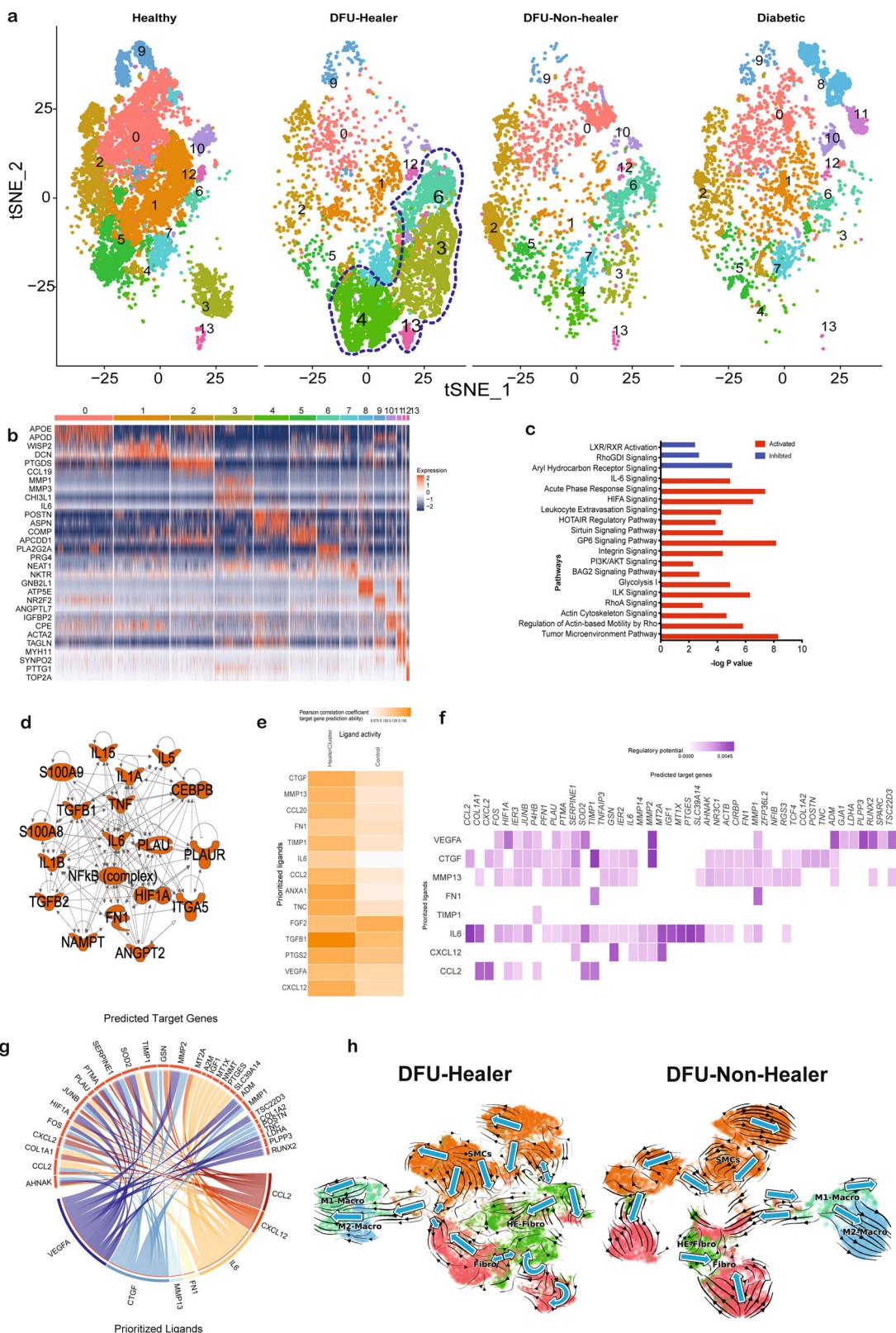

A heatmap displaying the connection between these key ligands that are expressed by HE-Fibro sub-cluster 3 (rows) and marker genes for the HE-Fibro sub-clusters (3, 4, 6, and 13) (columns) is shown in Fig. 5f and Supplementary Fig. 12. For example, *CCL2* overexpression in the "receiver" HE-Fibro sub-clusters can be strongly predicted by the presence of *IL6* in the "sender" HE-Fibro sub-cluster, indicating possible molecular interaction between them. The circos plot shows association between ligands from "sender" cells, sub-cluster 3 (lower hemicircle), and DEGs in "receiver" sub-clusters 4, 6, and 13 (upper hemicircle) (Fig. 5g, Supplementary Fig. 13). The analysis identified *IL6*, *MMP13*, *CCL2*, *CXCL12*, *CTGF*, *TIMP1*, and *VEGFA* as key regulatory ligands in the HE-Fibro sub-cluster 3, altering the expression of downstream target genes in the HE-Fibro sub-cluster 4, 6, and 13.

**Fig. 5 Identification and characterization of distinct subpopulations of fibroblasts with specific gene signature associated with healing DFUs.**
**a** t-distributed Stochastic Neighbor Embedding (t-SNE) analysis depicting 14 sub-clusters of fibroblasts. The sub-clusters enriched in DFU-Healers are marked with lasso. **b** Heatmap showing the top highly expressed genes (red) in sub-clusters. **c** Selected biological pathways that are significantly (*P* value <0.01) affected in the healing enriched fibroblasts. The directionality of each pathway is depicted using a pseudo color (red for activated, blue for inhibited). **d** Regulators that are significantly activated in the healing enriched fibroblasts. **e** Heatmap showing the Pearson correlation between ligands from 'sender' sub-cluster 3 and target gene expression in 'healer fibroblasts, i.e., the other HE-Fibro sub-clusters 4, 6, and 13 (left column), and 'control' fibroblasts sub-clusters 0, 2, and 5 (right column). A darker orange color indicates a higher Pearson correlation between the ligand and gene expression within the receiver cell population. **f** This heatmap of select ligands expressed by HE-Fibro sub-cluster 3 (rows) to regulate the genes which are differentially expressed by the 'healer' fibroblasts (columns). Well-established ligand-target gene interactions shown with a darker shade of purple. **g** Circos plot displaying the association between ligands expressed in the sub-cluster 3 (bottom semi-circle) with their targeted differentially expressed genes in sub-clusters 4, 6, and 13. **h** RNA Velocity plots for DFU-Healer and DFU-Non-healer subsets; black streamline arrows represent predicted direction of cell state change and trajectories. Larger blue arrows represent overall velocity for each area of the UMAP.

These identified ligands and their downstream targets might be responsible for the healing associated phenotype of HE-Fibro. Based on enrichment and specific regulatory interaction among HE-Fibro sub-clusters in DFU-Healers, we postulate that their role consists of creating a beneficial physiological environment for accelerated DFU healing.

**RNA velocity analysis predicts differentiation of HE-Fibro to other fibroblasts and SMCs.** To gauge the transcriptional dynamics of cell types of interest, we performed in silico trajectory analysis by computing the RNA velocity of Fibro, HE-Fibro, SMC, and M1- and M2-Macro cell populations in DFUs of Healers and Non-healers. We observed that the DFU-Healers cells were more fluid and interconnected, with distinct thick bridges between SMC and HE-Fibro or Fibro, reflecting potential transdifferentiation events. The analysis also predicted dual transdifferentiation of HE-Fibro toward SMCs and fibroblasts, as well as transdifferentiation of SMCs to macrophages.

Conversely, the DFU-Non-healers displayed more clearly defined branches of cellular trajectory and weak differentiation bridges among the cell types (Fig. 5h). Further, predicting the individual cells latent times revealed universally lower latent times for Healers cell types as compared to Non-healers cell types (Supplementary Fig. 14), thereby substantiating our hypothesis that Non-healers are stuck and fail to progress in the wound healing cascade. The lower latent times predicted for cell types from Healers indicate the likelihood that these cells are in earlier stages of differentiation and more stem cell-like.

**HE-Fibro display enrichment at the early recovery time point in acute wounds.** Our single-cell data identified significant enrichment of HE-Fibro and M1-Macro in the DFUs that eventually healed (Fig. 4b, c). To explore this enrichment in sequential wound samples during the healing process, we performed enrichment analysis in a previously published microarray dataset of acute wound healing[61] by calculating a score based on the HE-Fibro and M1-Macro gene signatures. The analysis revealed significant enrichment of HE-Fibro (Adj. *P*-Value = <0.0001) and M1-Macro (Adj. *P*-Value = <0.0001) at Days 4–8 post-injury samples in comparison to healthy unwounded skin (Supplementary Fig. 15), which correspond to the inflammation and early proliferation stages of healing. We did not observe enrichment of HE-Fibro and M1-Macro during the later phases of wound healing. These results provide further evidence that DFU-Healers exhibit a more acute-like response and have advanced forward in the wound healing process. Thus, enrichment of HE-Fibro and M1-Macro during the initial phases appears crucial for successful wound healing.

**Spatial transcriptomics and immunohistochemistry further elucidate gene expression patterns in healing and non-healing DFUs.** We subsequently selected well-defined surgically excised DFU sections from healers and non-healers for additional characterization. They both displayed blood vessel proliferation and chronic inflammatory cell infiltrates predominantly with perivascular distribution (Fig. 6a, c). We stained for inflammatory fibroblast markers, CHI3L1 and TIMP1, together with panfibroblast marker fibroblast activation protein (FAP) and discovered elevated numbers of triple-positive cells within the ulcer area of healing DFUs, with the cells forming dense aggregates (Fig. 6d). However, in the non-healing ulcers these cells were far fewer and scarcely distributed (Fig. 6b). We also evaluated gene expression using a spatial transcriptomics approach. The GeoMx® platform enables spatial, high-plex quantitation of gene expression in tissue through the use of in situ hybridization (ISH) probes that target mRNA in tissue; attached to the probes are photocleavable and indexed oligonucleotides that can be liberated via UV light and counted with an Illumina® sequencer. Regions of interest (ROIs) were chosen after staining for immune cell marker CD45, vasculature marker αSMA, and epithelial marker pan-Cytokeratin along with nuclear counterstain DAPI, to represent areas within the ulcer, at the edge of ulcer and adjacent non-injured tissue (Fig. 6a, c and Supplementary Fig. 16a, b). Hierarchical clustering analysis of representative healing and non-healing specimens revealed dissimilar gene expression profiles according to location within the sample: ROIs at similar dermal depth grouped together. (Fig. 6e, f). The non-healing ulcer ROI was particularly distinct from neighboring ROIs (Fig. 6e), while the healing ulcer ROIs appeared more transcriptionally similar (Fig. 6f). Focusing on the ulcer localized ROIs, DE analysis from two Healers (9 ROIs in total) and two Non-healers (4 ROIs in total) showed 148 genes upregulated in Healers and 57 in Non-healers (Fig. 6g and Supplementary Fig. 16c–f for additional DFUs). Among the most notable ones, HE-Fibro marker *PLA2G2A* and M1 macrophage marker *FOS* were overexpressed in Healers (Fig. 6h, i), while M2 macrophage markers *TYMP* and *ANXA1* were upregulated in Non-healers (Fig. 6j, k). Finally, gene ontology (GO) enrichment analysis unveiled cellular response to TNF as top biological function activated in healing ulcers and myeloid leukocyte migration in non-healing ulcers (Fig. 6l). Taken together, these findings verify our previous observations at the protein level and specify the location and functional roles of cell types reported in our scRNASeq dataset.

To further validate the finding based on spatial profiling that HE-Fibro mainly form niches in the wound bed to promote wound healing, we performed scRNASeq analysis on multiple samples from the same patient. ScRNASeq analysis was performed on skin specimens of the same patient from three different sites: wound bed, wound edge, and non-wound excess

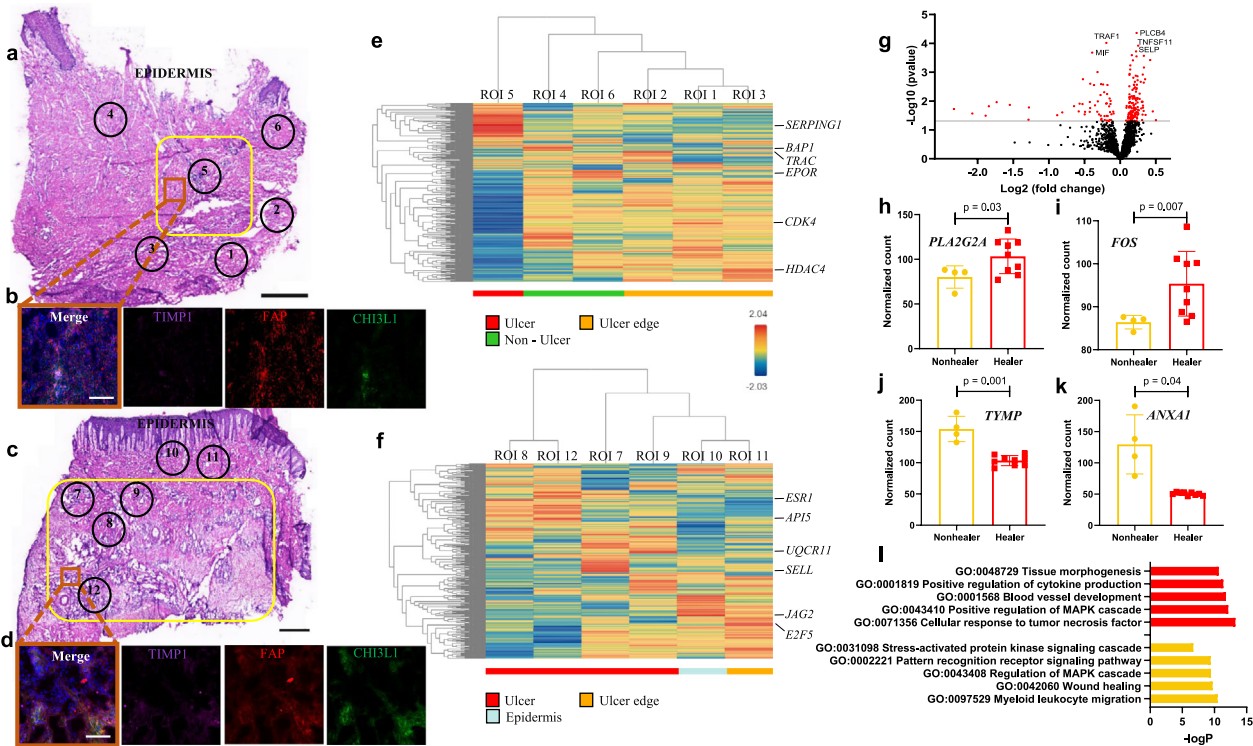

**Fig. 6 Exploring the spatial transcriptome of DFU-Healers and DFU-Non-healers. a, c** Representative H&E-stained sections from a (**a**) non-healing and (**c**) healing DFU. Yellow box demarcates the ulcer area and numbered circles the ROIs selected for sequencing. **b, d** Immunofluorescence staining for HE-Fibro markers TIMP1 (purple), CHI3L1 (green), and pan-fibroblast marker FAP (red) performed on a serial section from the same sample. DAPI was used for nuclear counterstain. The location of the image capture is noted with an orange box on (**a**) and (**c**). **e, f** Hierarchical clustering analysis heatmaps depict the transcriptomic profiles of the selected ROIs. The most highly expressed gene per ROI is highlighted. ROIs were annotated based on their location as Ulcer (red), Non-Ulcer (green), Ulcer edge (orange), and Epidermis (light blue). Expression levels are shown according to the gradient middle right (blue low to red high). **g** Volcano plot showing DE analysis results from ROIs within the ulcer in Healers (2 patients, 9 ROIs) vs Non-healers (2 patients, 4 ROIs). Each dot represents a gene, with red ones being above the significance threshold. The top five genes are highlighted. **h–k** Selected notable genes upregulated in Healers (**h**, **i**) and Non-healers (**j**, **k**). Data represent mean ± SD from $n = 4$ ulcer ROIs of 2 Non-healers and $n = 9$ ulcer ROIs of 2 Healers. Two-tailed unpaired t-test with Benjamini–Hochberg procedure for adjusted p-values was used to calculate p-values. **l** GO analysis for biological processes enriched in Healers (top, red) and Non-healers (bottom, yellow). Stainings were performed three times with two biologically independent patient samples per group. Scale bars are 1 mm in (**a**, **c**) and 100 μm in (**b**, **d**).

skin from a pressure sore excision (Supplementary Figs. 17 and 18). The unsupervised analysis and cellular annotation revealed that HE-Fibro were enriched in the wound bed, but not in the wound edge and non-wounded samples (Supplementary Fig. 18a). This unique wound bed enriched cell cluster exhibited significantly higher expression of HE-Fibro-associated genes like *IL6*, *TNFAIP6*, *MMP1*, and *CHI3L1* (Supplementary Fig. 18c). The absence of any other fibroblast cluster for the wound bed sample suggests that HE-Fibro originate from "normal" fibroblasts. These results further affirm an association of HE-Fibro with the wound healing process in an additional type of chronic wound and point toward heterogeneity of fibroblasts across different regions of ulcers.

**Induced overexpression of CHI3L1 in dermal fibroblasts influences cell behavior.** To explore the effects of inflammatory marker genes expression in vitro, we selected one of the top enriched genes *CHI3L1* and generated dermal fibroblast cell lines transduced with lentiviral vectors overexpressing *CHI3L1* (CHI3L1-OE) or a control sequence (CTRL). Western blotting demonstrated a complete lack of expression in untreated cells and RT-qPCR analyses confirmed a significant upregulation of CHI3L1 with construct 2 (Supplementary Fig. 19a, b), which we selected for further experiments. In adhesion assays, more CHI3L1-OE cells attached to fibronectin-coated surfaces

compared to CTRL (Supplementary Fig. 19c, d), while diminished migration was observed in scratch wound experiments (Supplementary Fig. 19e, f). Altogether, these findings shed light on the potential functional roles of the HE-Fibro, indicating that they possess enhanced adherent and decreased migratory capacities and suggest that they are firmly anchored on the ECM and mediate healing through secretion of molecules.

## Discussion

In this study, we performed large-scale unbiased scRNASeq to accurately and systematically profile patients with healing and non-healing DFUs, together with healthy non-DM subjects, and DM patients without DFUs, as controls. For a subset of patients, we also characterized forearm biopsies and PBMCs to evaluate any potential systemic effects of DM in presence of DFUs. To the best of our knowledge, we were the first groups to employ this approach in DFU samples[8], and we have now substantially expanded the number of cells sequenced, and incorporated state-of-the-art techniques like spatial transcriptomics, in order to gain novel insights into the transcriptomic landscape of DFU healing.

We identified a hitherto unreported fibroblast cell type associated with healing and expressing multiple immune and ECM remodeling-related genes. We then corroborated the results at the protein level and with the additional sequencing modality of spatial transcriptomics, demonstrating their localization within

the ulcer area. It has become increasingly apparent that dermal fibroblasts are a diverse and highly heterogeneous population with different functional roles in wound healing[62–66]. Fibroblasts at sites of inflammation, for instance within tertiary lymphoid structures, have been shown to acquire immune cell features[67], while in murine wounds' granulation tissue, a large proportion of fibroblasts is of myeloid cell origin[68]. A number of studies have also emphasized the interactions between fibroblasts and monocytes or macrophages in the context of inflammation and wound healing, implying a reciprocal relationship[69–71]. Our data suggest that specific fibroblast subtypes are key players in healing of DFUs and targeting them could be a therapeutic option.

Mapping the immune landscape of Healers and Non-healers revealed the presence of more M1 macrophages in Healers and M2 in Non-healers, as well as higher numbers of naïve and central memory T-cells in Healers, as opposed to more NK and NKT cells in Non-healers. Accumulating evidence suggests that a favorable outcome in wound healing is contingent on a highly regulated balance of macrophage polarized states[72]. The presence of more M2 macrophages does not necessarily equate better healing, as wound repair studies have shown delayed healing in diabetic[73] or wild-type[74] mice treated with M2 macrophages. DFUs are most probably populated both by tissue-resident macrophages that have differentiated from bone marrow-derived monocytes with a minimal contribution of yolk sac originating macrophages[75] as well as peripherally recruited monocytes responding to inflammatory cues and similarly differentiating to macrophages at the injury site[76]. To disentangle the admixture of macrophage subsets in DFUs and fully characterize their origins and repopulation dynamics, definitive future studies are required in human skin as has been accomplished in other organs[77].

Impairment in the recruitment of macrophages and neutrophils in DFUs was recently demonstrated[78]. A dysregulation in the differentiation of peripheral blood-derived T cells and diminished T-cell receptor repertoire diversity has been previously reported for DFU patients[79]. The majority of T-lymphocytes in our study originated from the blood samples, while macrophages were mostly located at the foot. Interestingly, in PBMCs of healers, inflammation pathways were mostly inhibited. These findings underline fundamental differences between systemic inflammation and the local wound inflammatory milieu. Overall, our results provide further evidence to support the claim that localized activated inflammatory response is required to surmount the chronic inflammation in DFUs, and progress to the next phases of wound healing[80,81], while, conversely, inhibition of inflammatory processes at the systemic level appears beneficial for healing. Trajectory analyses and comparison with acute wounds' gene expression demonstrated that DFU-Healers had advanced forward in the wound healing process and therefore the observed differences are not a distinct new pattern of healing but rather the difference between a dysregulated chronic inflammatory environment in DFU-Non-healers and wounds that have progressed to the first (inflammatory) or early second (proliferative) phases of healing in DFU-Healers.

Future longitudinal studies interrogating DFU samples collected from the same patient over multiple time points in the course of wound repair[82] could help build a map of the diabetic wound healing timeline. However, considering how technically challenging it is to consistently achieve high enough quality in debridement samples for next-generation sequencing[83,84], single-nucleus sequencing could be an alternative and complementary approach to offer single-cell resolution without relying on highly viable single-cell suspensions[85,86].

In summary, we present a comprehensive characterization of the DFU ecosystem and report novel cell types and interactions. Our dataset will be a valuable resource for diabetes, dermatology, and wound healing research, and can serve as the baseline for designing in vitro and in vivo experiments for the assessment of therapeutic interventions focusing on one or more cell types. Future studies utilizing pre-enrichment via flow or magnetic cell sorting could further characterize specific populations and lead to the discovery of rare cells.

## Methods

**Subjects**. Our study includes non-DM patients ($n = 10$) who underwent foot surgery for various reasons, such as hallux valgus correction, as the healthy controls, and DM patients without foot ulceration ($n = 6$) who had similar foot surgery. Discarded skin specimens from the dorsum of the foot were collected for analysis. We also enrolled DM patients with plantar foot ulceration (DFU) ($n = 11$), who underwent surgical resection of the ulcer, providing sufficient wound and peri-wound tissue for analysis. Subjects with any conditions, other than DM, or medications that could affect wound healing were excluded from the study. Four non-DM subjects, two DM patients with no DFU, and five DM patients with DFU (Healers; $n = 3$, Non-healers; $n = 2$) provided two 3-mm forearm skin biopsies and 20 ml of blood, from which PBMCs were isolated, within 1 week of the foot surgery. DFU patients were followed for 12 weeks post-surgery and were divided into two subgroups: those who healed their ulcers and those who failed to heal them (Healers; $n = 7$, Non-healers; $n = 4$). Supplementary Dataset 4 includes clinical details of the subjects included in the study. There were no major differences among the main biological characteristics of the studied groups (Supplementary Table 2). All patients were enrolled and followed at the Joslin-Beth Israel Deaconess Foot Center, Boston, MA, and the study was approved by the Beth Israel Deaconess Medical Center IRB (Reference number 2018P000581). For the scRNASeq analysis of spatially separated samples, multiple samples were collected from an ischial pressure sore of one patient at the Yale Plastic and Reconstructive Surgery – Wound Center, New Haven, CT (collected under IRB approval 1609018360). Informed consent was obtained from all study participants at Beth Israel Deaconess Medical Center and Yale. For comparative analysis of enrichment of various healing associated cell types in sequential wound healing samples, we downloaded microarray burn study data of 5 patients from the ArrayExpress database, accession ID E-MTAB-1323.

**PBMCs isolation**. PBMCs were separated using Ficoll-Paque density gradient fractionation, as previously described[87], and cryopreserved in freshly prepared freezing media (90% FBS and 10% DMSO).

**Single-cell preparation from skin samples**. Skin specimens were kept in sterile PBS on ice until processing, normally within 3 h post-surgery. The skin was cleaned by sequentially immersing in 10% Betadine, 70% ethanol, and PBS for 1 min at a time. Then it was incubated in 5 mg/ml Dispase II (Thermo Fisher Scientific, 17105041) in HBSS (STEMCELL Technologies, 37150) overnight at 4 °C. The next day, the epidermis was peeled off using forceps, and the tissue was finely minced with a No. 10 disposable scalpel. The skin pieces were then placed in an enzyme cocktail consisting of 3.3 mg/ml Collagenase-P (Roche, 11249002001), 3.3 mg/ml Dispase II, and 1.5 mg/ml DNase I (STEMCELL Technologies, 07470) in 0.25% Trypsin-EDTA (Thermo Fisher Scientific, 25200072) and incubated for 90 min at 37 °C with constant shaking, using glass pipettes for trituration every 20 min. Enzymes were then inactivated with the addition of complete DMEM (+10%FBS, +1% Pen/Strep). The single-cell suspension was passed through 70 and 40 μm cell strainers and centrifuged for 10 min, 500 × g at 4 °C. For red blood cell (RBC) lysis, ACK buffer (Lonza, 10-548E) was added. The process resulted in highly viable, typically >90%, single-cell suspensions. For immediate single-cell capture, the cells were resuspended in 0.04% Ultra-Pure BSA in PBS (Thermo Fisher Scientific) and concentration was adjusted to 1000 cells/μl. If not processing for scRNASeq immediately, the cells were cryopreserved in freshly prepared freezing media (90% FBS and 10% DMSO).

**Single-cell RNA sequencing**. The single-cell preparations of the foot, forearm, and PBMC samples were used fresh or after thawing of viably frozen samples with final resuspension in PBS with 1% BSA. A droplet-based ultra-high throughput scRNASeq system was utilized to capture single cells along with uniquely barcoded primer beads together in tiny droplets, enabling large-scale parallel single-cell gene expression studies. The gene expression (GEX) libraries were prepared using the Chromium 3'V2/3 reagent kits (10x Genomics, 120237 and 1000075). Briefly, gel bead-in-emulsions (GEMs) were generated and barcoded by loading single-cell suspensions along with gel beads and reverse transcription (RT) master mix in 10x Genomics Single cell chip (A chip kit, 120236; B chip kit, 1000153) and running on the chromium controller (10x Genomics, 110211). Following RT, the cDNA was amplified and used to generate GEX libraries. The cDNA and GEX libraries were quantified using Qubit 3.0 fluorometer (Life Technologies, 15387293), and quality was assessed using HS DNA chips (Agilent technologies, 5067-4627) with 2100 Bioanalyzer (Agilent Technologies, G2939BA). Sequencing was performed using massively parallel sequencing on the Novaseq S4 platform (Illumina). We produced

~40,000–50,000 reads per cell capturing the expression of ~1000–2000 transcripts per cell.

**Data processing and analysis**. Raw scRNASeq data was demultiplexed, aligned to the reference human genome (Hg38), and processed for single-cell gene counting using the Cell Ranger Software from 10X Genomics Inc. The single-cell count data was normalized using the SCTransform algorithm in Seurat v3.0 Bioconductor package[88] that uses regularized negative binomial models for normalizing sparse single-cell data. The normalized expression profiles of the samples were merged, and undergone quality control, pre-processing, unsupervised and supervised analysis using various R and Bioconductor packages. The quality filtering on scRNAseq data was performed by multiple filtering parameters including >50% of mitochondrial genes, cells expressing the lower number of genes (<200 genes), and genes only uniquely expressed in <3 cells.

The unsupervised analysis using principal component analysis (PCA) was performed on variable genes to identify principal components, which captured the most variance across the samples. These principal components were used as an input for Uniform Manifold Approximation and Projection (UMAP) analysis[89] to determine the overall relationship among the cells. Cells with similar transcriptome profiles clustered together, and the clusters were subsequently annotated to different cell types based on the expression of specific well-established cell marker transcripts. Comparative analysis of the single-cell landscape of healing and non-healing DFUs, along with healthy non-DM subjects and non-DFU DM patients as controls, was performed using split UMAP plots, for determining heterogeneity (based on clusters of cells) and abundance of cell types. The significance testing change in abundance of cell types across clinical groups was performed either using one-way ANOVA or Welch's $t$-test ($p$-value < 0.05). Similar analysis was also performed for the 3 different anatomical sites separately from where the samples were collected, i.e., foot, forearm, and peripheral blood. To further characterize cell type-specific differences among clinical groups, we performed comparative analyses using multiple tests corrected non-parametric Wilcoxon Rank Sum test ($P$ Adjusted value = 0.01, Fold Change = 1.2) on individual cell types like fibroblasts, keratinocytes, T-lymphocytes, natural killer cells, monocytes, macrophages, mast cells, B-lymphocytes, plasma cells, and dendritic cells.

**Pathways and systems biology analysis**. To precisely characterize the cell types and understand the molecular mechanism of wound healing, we performed pathways enrichment and systems biology analysis. The analysis was performed on transcripts that were significantly dysregulated in the specific cells by comparing healed vs non-healed samples. Pathways and systems biology analysis was performed using the Ingenuity Pathway Analysis software package (IPA 9.0) (Qiagen). A detailed description of IPA is available at the Ingenuity Systems' website (http://www.ingenuity.com). Systems biology analysis was performed by analyzing the upstream transcriptional regulators. The regulatory analysis helps in identifying significantly activated or inhibited transcriptional regulators based on upregulation or downregulation of its target genes. The significance of transcriptional regulators activation/inhibition was determined using one-tailed Fisher's exact test. The regulators with a $p$-value <0.01 and absolute z-score 2 were considered statistically significant.

**Ligand and receptor-based cell interaction analysis**. NicheNetR[60] was used to identify ligands produced by Healer-specific fibroblasts, which could uniquely regulate other healer-specific fibroblasts. NicheNetR uses a prior model of ligand-target interactions derived from a meta-analysis of multiple sources to identify ligands that may explain expression differences in a given set. In this workflow, cells are classified as either senders or receivers. The expression of sender cells is used to identify possible ligands, while the receiver cells are used to generate a gene set. In this case, Cluster 3 with overexpression of *MMP1, MMP3, CHI3L1, CCL20*, and *TIMP1* from the Healer-specific fibroblasts was treated as a sender cluster, while other Healer-specific clusters (Fig. 5; clusters 4, 6, 13) or non-specific clusters (Fig. 5; 0, 2, 5) were treated as receivers. The gene sets used were the markers differentially expressed between DFU-Healers and DFU-Non-healer samples within the receiver subsets. The top markers are combined with NicheNetR's ligand-target weights to compute the Pearson correlation coefficient between ligands and expression changes in the receiver subset. A high Pearson correlation coefficient between ligand and target gene set indicates that expression of ligand might be responsible for expression differences. For a ligand to be considered for interaction analysis, it must be expressed in at least 5% of the sender cell population, and its corresponding receptor must be expressed in 5% of the receiver cell population.

**Cellular trajectory and differentiation state analysis**. To measure the transcriptional dynamics and characterize differentiation process based on single-cell data from DFU healers and non-healers, we performed RNA velocity analysis using the Velocyto[90] and scVelo algorithms[91]. We performed RNA velocity analysis on HE-Fibro, Fibro, SMCs, M1-Macro, and M2-Macro cells in the DFU-Healer and DFU-Non-healer subsets of the foot cells. We generated spliced and unspliced counts using the Velocyto package and merged the data from all patients. This merged dataset was subsetted to retain only HE-Fibro, Fibro, SMCs, M1-Macro,

and M2-Macro cells from foot samples as they showed significant association with the wound healing process. Velocity streams and inference of root cells were generated using scVelo version 0.2.3[91]. This python package uses steady state and dynamic models to predict cell trajectory and latent time. The RNA velocities were projected onto a computed UMAP for each subset; the streamline velocity vectors represent directions and flow of estimated trajectory and differentiation of the cells. The dynamic model also predicts the latent time of the cells, which represents the cell's position in a biological process.

**Enrichment of HE-Fibro and Macrophages gene signatures in the temporal healing data from acute burn wound study**. To assess the enrichment of HE-Fibro and M1 Macrophages in the sequential wound healing samples, we performed external validation using microarray data from a temporal acute burn wound study. The raw data downloaded from the ArrayExpress database accession number E-MTAB-1323 contain gene expression data of skin from 5 patients on 6 different time points, pre- and post-burn injury. The R packages beadarray[92] and limma[93] were used to pre-process, normalize, and analyze gene expression data. Gene set enrichment analysis (GSEA) was performed using the GSVA R package[94] to compare enrichment of HE-Fibro and M1-Macro gene signatures. These signatures are subsets of differentially expressed genes comparing DFU-Healer and DFU-Non-healer subsets. The HE-Fibro gene signature used in GSEA analysis consists of *PLA2GA, MMP1, CHI3L1, TIMP1, SFRP4, FTH1, FN1, MT2A, LUM, CHI3L2, MMP13, HIF1A, CCL20, TPM2, ASPN, MMP3, TNFAIP6*, and *IL6*. The M1-Macro gene signature consists of *IL1B, S100A8, VCAN, BCL2A1, LYZ, S100A9, TIMP1, C15orf48, SRGN, NFKBIA, BTG1, NAMPT, PLAUR, SAT1, ID2, TYMP, SLC2A3, SERPINA1, CXCL8*, and *SOD2*.

After calculating enrichment scores of the signatures at each time point for individual subjects, average enrichment scores and standard deviation of each time point were calculated. Further statistical significance of changes in the enrichment score was determined by one-way mixed-effects ANOVA with Bonferroni corrections.

**Immunofluorescence staining and imaging**. For confirmation of HE-Fibro presence, 5-μm-thick frozen sections from healing and non-healing DFUs were fixed in 80% ice-cold acetone for 10 min, blocked with 5% donkey serum in 0.2% PBS-Tween for 30 min at room temperature, and incubated overnight in a humidified chamber at 4 °C with primary antibodies: mouse monoclonal anti-FAP (1:50, clone F11-24, sc-65398, Santa Cruz Biotechnology), rabbit polyclonal anti-CHI3L1 (1:100, ab77528, Abcam) and goat polyclonal anti-TIMP1 (1:100, AF970, R&D Systems). Alexa Fluor donkey anti-rabbit 488-, anti-mouse 594- and anti-goat 647-conjugated secondary antibodies (1:1000 ab150061, 1:500 ab150112, and 1:1000 ab150131, respectively, all Abcam) were added the next day for 1 h at room temperature. 2-(4-amidinophenyl)-1H-indole-6-carboxamidine (DAPI) was included for nuclear counterstaining. TrueVIEW Autofluorescence Quenching Kit (Vector Labs, SP-8400) treatment was employed to enhance staining. Tissue sections were mounted in ProLong Gold Antifade (Thermo Fisher Scientific, P36930) and visualized with a Zeiss LSM 880 (Carl Zeiss) inverted confocal microscope and images processed with ZEN 2011 (Carl Zeiss) and ImageJ/FIJI (NIH) software packages.

For validation of macrophages and most significant pathways, paraffin-embedded immunofluorescent staining of healing and non-healing DFUs was performed. 5-μm-thick sections were deparaffinized, rehydrated and antigen retrieval was achieved with citrate buffer pH 6.0 in a pressure cooker for 20 min. The sections were then blocked with 5% donkey serum in 0.2% PBS-Tween for 1 h at room temperature and incubated overnight in a humidified chamber at 4 °C with primary antibodies: goat polyclonal anti-IL17 (1:100, AF-317-NA, R&D Systems), goat polyclonal anti-TNFA (1:50, AF-410-NA, R&D Systems), rabbit monoclonal anti-HIF1A (1:100, clone EP1215Y, ab51608, Abcam), rabbit polyclonal anti-IL6 (1:100, ab6672, Abcam), goat polyclonal anti-IL1B (1:50, AF-201-NA, R&D Systems), goat polyclonal anti-S100A8 (1:100, AF3059, R&D Systems), mouse monoclonal anti-DAB2 (1:50, clone E-11, sc-136964, Santa Cruz Biotechnology), rabbit monoclonal anti-CD68 (1:100, clone EPR20545, ab213363, Abcam), mouse monoclonal anti-VIM (1:100, clone V9, MAB3400, Sigma), mouse monoclonal anti-CD163 (1:20, clone GHI/61, sc-20066, Santa Cruz Biotechnology). Appropriate admixtures of Alexa Fluor donkey anti-rabbit 488- (1:1000, ab150061) and 594- (1:500, ab150064), anti-mouse 488- (1:1000, ab150109) and 647- (1:1000, ab150107), and anti-goat 594- (1:500, ab150132) and 647- (1:1000, ab150131) conjugated secondary antibodies, all from Abcam, were added the next day for 1 h at room temperature. DAPI was included for nuclear staining. Sections were quenched for 5 min using the TrueView Autofluorescence Quenching kit to decrease background (Vector Laboratories) and covered with anti-fade mounting medium. Images were obtained at ×20 magnification with an Axio Imager A2 upright microscope using Zen Blue edition software (Zeiss).

CD68/Vimentin (VIM) were used as guide stains to capture images within the ulcers with similar abundance of positively stained cells. Quantification was performed on ImageJ/FIJI by counting the number of double-positive CD68 cells with respective macrophage polarization markers (DAB2, S100A8, IL1B, and CD163) and dividing by the area of the tissue for normalization; counting the number of IL17 positive cells per area; computing the percentage of stained area for TNFA, IL6, and HIF1A. Two measurements were averaged per sample.

**Spatial transcriptomics**. The spatial transcriptome profiling was performed using NanoString's GeoMx Digital Spatial profiling platform on unfixed frozen 5-μm tissue sections. Samples were processed as follows: (1) 10% neutral buffered formalin (NBF) fixation overnight, (2) target retrieval (1X Tris EDTA, pH 9.0 for 20 min), (3) proteinase K digestion (1 μg/mL for 15 min), (4) post-fixation (10% NBF, Tris-glycine stop buffer), (5) in situ hybridization overnight with the GeoMx Cancer Transcriptome Atlas probe panel (1800-plex), (6) stringent washes (50:50 formamide/4X SSC), and (7) fluorescent antibody/marker (aSMA, 1:100, Clone: 1A4, Abcam; CD45, 1:100, Clone: 2B11 + PD7/26, Novus; PanCK, 1:50, Clone: AE1/AE3, Novus) incubation, 1 h at room temperature. Sections were then loaded onto the GeoMx® Digital Spatial Profiler (Nanostring, GMX-DSP). For profiling, circular regions of interest (ROIs), ~500 μm in diameter, located within the ulcers or in neighboring non-ulcerated tissue were selected to include high concentrations of CD45+ immune cells in close proximity to vessels (αSMA + structures). After ROI selection, the GeoMx instrument illuminated each ROI separately with UV light to cleave, aspirate, and deposit the oligonucleotides from the hybridized ISH probes for downstream sequencing into a 96-well plate. Library preparation (PCR, AMPure bead purification) was performed, followed by paired-end sequencing with an Illumina NextSeq 550. Sequencing data (FASTQs) was then processed with a custom GeoMx NGS pipeline (DCCs) to be analyzed in part with the GeoMx Data Analysis Suite. Raw reads were processed for high quality with TrimGalore and FLASH[95]. Reads were then aligned to analyte barcode with Bowtie2[96]. PCR duplicates were discarded using UMI-tools with the Hamming distance set at three. Poorly performing probes were removed from analysis if they were outliers (Grubbs test) or had low counts relative to other probes targeting the same gene. Raw probe count data (up to 5 unique probes per gene) were condensed into gene level count data and normalized with the quartile 3 gene count value per ROI individually. Complete-linkage hierarchical clustering was performed on normalized counts and represented by heatmap using the R function pheatmap. Unpaired $t$-test with Benjamini–Hochberg procedure for adjusted $p$-values was used to calculate differentially expressed genes with a threshold $p < 0.05$. Significantly expressed genes were entered on Metascape (Version 3.5, http://metascape.org) for enrichment analysis with Gene Ontology (GO) Biological Processes (Version 2020-09-16). All genes in the human genome were used as the enrichment background. $P$-values were calculated based on cumulative hypergeometric distribution and Q-values were calculated using the Benjamini–Hochberg procedure for multiple testing. A term was considered overrepresented when $p < 0.01$, had a minimum count of 3 and an enrichment factor >1.5, which is the ratio between the observed counts and the counts expected by chance. Volcano plot and GO bar graphs were designed with Prism 8.4.2 (GraphPad).

**Cell culture**. Normal human dermal fibroblast cells (BJ CRL-2522) were obtained from ATCC and maintained in Eagle's minimum essential medium (EMEM) (ATCC, 30-2003), supplemented with 1% (v/v) penicillin/streptomycin (P/S) and 10% (v/v) fetal bovine serum (FBS) (Sigma-Aldrich, F1435). For passaging, cells at ~80% confluence were detached through a 5- to 10-min incubation with 0.05% Trypsin/EDTA and further resuspended in complete EMEM. Cells were then centrifuged at 1200 rpm for 5 min. The cells were replated at a concentration of 6000 cells/cm$^2$ and/or cryopreserved with 90% FBS and 10% DMSO freezing media. Cells were maintained in 95% O$_2$, 5% CO$_2$ at 37 °C and routinely tested for mycoplasma contamination (PromoKine, PK-CA91-1096).

**Transduction of fibroblasts with Precision LentiORF viral vectors**. Cells were seeded in 6-well culture plate at 150,000 cells per well and pre-incubated with 5 μg/ml polybrene for 10 min at 37 °C. Afterward, cells were incubated overnight with culture medium containing 5 μg/ml polybrene and the viral particles carrying the CHI3L1 gene (OHS5899-202624268, Horizon Discoveries) or the positive control viral particles (OHS5833) at a multiplicity of infection of 10. After removal of the particles containing medium, cells were incubated in culture medium with 10 μg/ml Blasticidin to positively select transduced cells. Transduction efficiency was evaluated with assessment of GFP expression, for both target and control constructs and RFP expression, for the control construct, on a K2 Cellometer (Nexcelom Bioscience) and with live-cell imaging on a Zeiss LSM 880 microscope.

**Real-time qPCR**. RNA was extracted from 100,000 cells using the miRNeasy Mini Kit (Qiagen, 217004). RNA quantification was done by using the Qubit RNA BR Assay kit (Cat. No. Q10210) and the Qubit 3 Fluorometer. cDNA was used at a concentration of 15 ng/ml from 1 μg of RNA and reverse transcribed with the miScript II RT kit (Cat No. 218161). RT-qPCR analysis was run for the samples using a miScript SYBR Green PCR Kit (Cat. No. 218073) on a Stratagene Mx3005P (Agilent Technologies). Housekeeping gene GAPDH primers were purchased from Qiagen (Cat. No. QT00079247) and CHI3L1 primers, were obtained from MGH-HMS primer bank with the following sequences: FW: 5′-GAA GAC TCT CTT GTC TGT CGG A-3′ and RV: 5′-AAT GGC GGT ACT GAC TTG ATG-3′. Data were normalized to the expression of GAPDH and were analyzed using the $2^{-\Delta\Delta CT}$ method.

**Western blotting**. 500,000 cells were centrifuged at $130 \times g$ for 5 min at 4 °C, washed with ice-cold PBS, and centrifuged again at $2400 \times g$ for 5 min at 4 °C.

The pellet was then resuspended in ice-cold RIPA buffer (Prod# 89901) supplemented with 10 μl/ml protease and phosphatase inhibitors (Prod# 78430 and 78420) and incubated for 15 min on ice with periodical pipetting and vortexing. Samples were then centrifuged at $14,000 \times g$ for 15 min at 4 °C and supernatants were collected. Protein concentration was measured using the Pierce™ BCA Protein Assay Kit (Cat. No. 23225). The protein samples were reduced by using a 6x Laemmli buffer, and boiled at 95 °C for 5 min. 30 μg of protein per sample was loaded into 12% SDS-PAGE gels and run at a constant 200 V for 40 min. The gel was washed with Tris-buffered saline-Tween 20 (TBST), and incubated in blotting buffer for 10 min. The transfer ran overnight in a cold room at a constant 90 mA. Once transfer was complete, the blot was washed in TBST, and blocked with 5% BSA for 1 h at room temperature. Blot was then incubated in 5% BSA with CHI3L1 (Abcam, ab77528 1:1000) or GAPDH (ab9485, 1:5000) antibodies for 1 h at room temperature. Afterward, the blot was washed in TBST, and incubated with a secondary antibody (ab205718 1:10,000) for 1 h at room temperature. Finally, a chemiluminescent substrate (Cat. #1705062) was added and the blot was visualized using the ChemiDoc™ Touch Imaging System (Bio-Rad). For stripping, the blot was washed in TBST and incubated in stripping buffer (Prod# 46430) for 45 min at room temperature. After stripping, the blot was washed with TBST, blocked with 5% BSA for 1 h at room temperature, and reprobed as previously described.

**Adhesion assay**. Transduced BJ cells were plated at 50,000 cells per well in 12-well plates pre-coated with 10 μg/ml human fibronectin (Prod# 33016-015) and incubated for 1 h at 37 °C. Afterward, cells were washed with PBS, fixed with 4% paraformaldehyde for 15 min, and stained with 0.05% crystal violet for 30 min at room temperature. Pictures of the adherent cells were taken on a Primo Vert inverted microscope (Carl Zeiss) with an Axiocam 105 camera. Pictures of two random fields were taken per well, for at least three wells per condition, and cells were counted using ImageJ/FIJI software. Three independent experiments were performed.

**Scratch assay**. Transduced BJ cells were plated at 50,000 cells per well in 24-well plates. The cell monolayer was scratched in a straight line using a 200 μl pipette tip. Debris were removed by washing once with media, then cells were incubated in medium supplemented with 5% FBS throughout the experiment. Images were taken immediately after the scratch, in 6 h, and in 12 h for four wells per condition. Scratch areas were analyzed using ImageJ/FIJI. Three independent experiments were performed.

**Reporting summary**. Further information on research design is available in the Nature Research Reporting Summary linked to this article.

## Data availability
Spatial transcriptomics and scRNASeq data have been submitted to NCBI's Gene Expression Omnibus (GEO) and are accessible through GEO accession numbers GSE166120 and GSE165816. Burn wound gene expression data were downloaded from ArrayExpress accession number E-MTAB-1323. An interactive data resource and analytical tool developed based on this DFU single-cell data are available online at https://bhasinlab.bmi.emory.edu/Diacomp. Source data are provided with this paper.

## Code availability
Code for data analyses is described in the "Methods" and is available from the corresponding authors upon request.

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

## Acknowledgements

This work was supported by the NIDDK-sponsored Diabetic Complications Consortium grant 5U24DK115255-04 (A.V. and M.B.). A.V. received funding from the National Rongxiang Xu Foundation. G.T. received a George and Marie Vergottis Foundation Postdoctoral Fellowship award.

## Author contributions

V.H., A.V., and M.B. obtained funding. H.C.H., V.H., A.V., and M.B. supervised the project. G.T., A.V., and M.B. designed the experiments. G.T., B.E.T., D.S., S.S.B., T.S.S., R.F.S., I.M., P.W., and A.L. conducted the experiments. G.T., B.E.T., D.S., H.L.M., W.J.R.P., B.D., W.P., A.K., I.V., S.S.B., A.V., and M.B. analyzed data. G.T., B.E.T., D.S., S.S.B., H.L.M., W.J.R.P., A.V., and M.B. wrote the manuscript.

## Competing interests

The authors declare no competing interests.

## Additional information

**Peer review information** *Nature Communications* thanks Anna Alemany, David Paik, Hongting Zheng and the other anonymous reviewer(s) for their contribution to the peer review this work. Peer reviewer reports are available.

