## [Peer Review File · Nature Communications]

Reviewers' Comments:

Reviewer #1:

Remarks to the Author:

This manuscript by Theocharidis et al. profiled up to 200k cells from foot and forearm skin biopsies as well as PBMCs from diabetic foot ulcer patients (n=11) and non-diabetes patients (n=10) to assess the molecular and cellular associations between diabetes and ulcer healing. The manuscript consists of high-quality characterization of single-cell and spatial transcriptomic analysis of various cell types involved including macrophages and "Healing Enriched" fibroblasts. While the manuscript is largely descriptive in nature, it provides a comprehensive and informative cellular atlas of human DFU patient tissue.

Minor points:

- Would the authors be able to provide basic clinical information of all patients from whom tissues were collected, e.g., age, racial/ethnic information, gender, and other pre-existing medical conditions most notably cardiometabolic parameters, as available? This can be shown in a supplemental table.
- In Figure 5G and H, the gene names in the heatmap and the Circos plot are difficult to read. Would there be a way to show a selected number of representative ligand-receptor pairs that authors would like to highlight the most in a more visible format?
- Have authors tried any of the available cellular trajectory and/or RNA velocity pipelines on Fibro/HE-Fibro-SMC populations to assess whether there is a transdifferentiation event among these cell types?

Reviewer #2:

Remarks to the Author:

The authors conducted scRNASeq in skin tissue and PBMCs from non-DM patients and DM patients with or without DFU, and found that a unique population of fibroblasts with high expression of genes related to ECM remodeling and immune/inflammation were over-represented in the samples from DFU-healers. Moreover, the healing associated fibroblasts were more like to be presented in wound/ulcer bed rather than wound edge or non-wounded skin. Except for fibroblasts, authors also revealed that more M1 macrophages and naïve and central memory T-cells were presented in healers, while more M2 cells and NK and NKT cells in non-healers.

Overall, the study is well-designed, the findings were interpreted appropriately, and like the authors said that their results would be valuable sources for other related researches. But I still have one question. As the wound healing is a complex process consisting of multiple overlapping stages, is it possible the differences of cell type/gene expression pattern between DFU-healers and non-healers were the same with the differences between the last stage of healing process and the former stages? In other words, DFU-non-healers might be stuck in certain stages of wound healing rather than present as a new pattern of healing. Please discuss this question.

Reviewer #3:

Remarks to the Author:

The authors made a profound effort to understand the microenvironment of healing and non-healing wounds at two different locations in diabetic foot ulcers using a single cell RNA seq-approach. They created a landscape of different cells potentially interacting within the environment of healing and non-healing diabetic wounds. They found uniquely enriched fibroblast and M1 macrophage populations in diabetic foot ulcers and associated them with better healing. This study provided comprehensive correlative data but still lacks mechanistic details and conclusive evidence that truly advances our understandings of the non-healing diabetic foot ulcers. I have few major concerns on this study:

- 1) It is very difficult to compare healing and non-healing diabetic foot ulcers only for one time point. Impaired healing may lag behind relatively when compared to healing wounds. Therefore, a time kinetic would be important. From previous publications there is evidence that M1

macrophages play a role in removal of debris and counteract infections at the wound site. A switch to the M2 subtype of macrophages is essential for wound healing. This switch does not occur in murine diabetic wounds (Mirza R., Koh T. J. (2011). Dysregulation of monocyte/macrophage phenotype in wounds of diabetic mice and human diabetic foot ulcers. *Cytokine* 56, 256–264. 10.1016/j.cyto.2011.06.016; Mirza, R. E et al (2013) Blocking interleukin-1 β induces a healing-associated wound macrophage phenotype and improves healing in type 2 diabetes. *Diabetes* 62: 2579–2587; Lim et al. (2015) relative expression of proinflammatory and antiinflammatory genes reveals differences between healing and non healing human chronic diabetic foot ulcers. *J Invest Dermatol* 135:1700-1703.). This example demonstrates the importance of a time kinetic for single cell seq analysis of diabetic healing and non-healing wounds. The analysis of only one time point is prone to misinterpretation.

2) Authors should analyse foot healing over a time period in particular enrichment kinetics of different cells population and associated gene signatures. Analysis of wound biopsies from diabetic and acute wounds collected at different time points would solidify their conclusions. It would be interesting to see whether diabetic healing wound closely overlap with healthy acute human wounds in terms of fibroblasts, keratinocytes, T cells and macrophages recruitment and transcriptome. It is also not clear what drives accumulation of uniquely enriched fibroblasts or M1 macrophages in healing foot ulcers and whether healing fibroblasts or immune cells influence each others recruitment in a paracrine manner.

3) Authors should characterize M1 and M2 macrophages in more detail and experimentally explain how accumulated M2 macrophages oppose healing of diabetic foot. What is the origin or source of these distinct cells type whether they proliferate or differentiate locally or were recruited from distant organs. Some aspects of this manuscript are not entirely novel, as some of the transcriptional findings of human diabetic foot ulcers and dysregulated inflammation has already been reported by Sawaya et. al. (*Nat Commun* . 2020 Sep 16;11(1):4678.).

4) Authors should provide more data on how activated IL-6 signalling drives healing of diabetic wound, as this previously have been reported to promote local inflammation and prevent wound healing.

5) Validation of selected biological pathways shown in Figure 4 and Figure 5 in diabetic foot ulcers as well as diabetic and healthy age/gender matched foot pad skin is required at protein level to support findings at transcriptional levels. Author are recommended to causally validate some of the key findings in diabetic mice wound healing model using genetic or pharmacological approaches.

6) More detailed protein analysis including chemokine, cytokines and their major regulators would help to better understand the wound microenvironment. Multiplex immunostaining of different fibroblasts and immune cells in conjunction with highlighted pathways proteins at different time points is required.

7) Authors should provide demographic details of the patient samples. (Duration of diabetes mellitus, wound size, HbA1c, comorbidities, age, medication). This would be best shown in a table.

Reviewer #4:

Remarks to the Author:

In this paper, Theocharidis et al investigate the diabetic wound healing microenvironment by comparing the transcriptomic profile of the PBMCs, foot and forearm in diabetic patients suffering from diabetic foot ulcers (DFU) that heal (healers), diabetic patients suffering from DFU that cannot heal (non-healers), diabetic patients with no DFU (DM patients), and non-diabetic patients (non-DM patients). The authors perform scRNA-seq analysis at different levels, first with all the cells sequenced together and later performing the analysis to PBMC samples only, to NK/NKT/T cells only, to cells from the foot only, and to fibroblasts only. With this approach, a fibroblast cell type associated with wound healing is reported, and its existence and localization is validated with spatial transcriptomics. In addition, differences in the population ratios of immune cells are also

detected between the different clinical groups.

The paper is well written and easy to follow. The findings are well justified and will be useful for investigators (and hopefully patients!) in the diabetic research field.

However, the data is presented in a very qualitative approach and it is hard to trust some of the reported enrichments in terms of cell type population fractions. I hope that by addressing the major comments below, the paper becomes more quantitative.

Major comments:

- In the methods, the authors state that the significance testing change in abundance of cell types across clinical groups was performed either using ANOVA or Welch's t-test. However, no p-values are reported in several barplots (e.g. Figure 1d), leaving it to the reader to assess/believe significant enrichments of cell types in the different clinical groups. It would be good to indicate enrichment p-values in the figures.

- In Figure 1d, erythros and plasma seem to be enriched in non-healers, but nothing is stated. Additionally, Mast cells also seem enriched in the controls. Is there any biological or technical reason for it?

- In the first section of the results, what do the authors mean by "orthogonal clusters"?

- Being a clinical study, I expect there to be batch effects because of patient of origin. Could the authors show whether HE-Fibros are equally present in all healer patients?

- How are the error bars in Figure 3b computed?

- What is the rationale to select cluster 3 as the "sender" population in the NicheNetR analysis perform in Figure 5?

- The authors perform scRNA-seq analysis on multiple samples from the same patient. Which clinical group does the patient belong to (I assume it is a healer)? Has this data been produced 3 weeks after previous scRNA-seq data? Can the authors extract any information about the population dynamics of the HE-fibro cell type (e.g. does their fraction increase/decrease during the course of wound healing?)

Minor comments:

- In the abstract, the authors use the acronyms PBMC and DFU without previous definition. This can be confusing for non-experts.

- Did the authors consider getting skin biopsies from the foot 12 weeks post-surgery in healers and non-healers to investigate dynamic changes in the wound microenvironment? Could this reveal some temporal differences in the establishment of fibroblast populations required for wound healing?

- The website (<https://bhasinlab.bmi.emory.edu/Diacomp/>) is not user-friendly: tick-boxes overlap with the UMAP's legend unless the user displays the project summary. When the project summary is displayed, tick-boxes are too far from the UMAP and the user needs to keep scrolling up and down. Can this be optimized? Additionally, can Supplementary Table 2 be made accessible via the website?

- What do the dashed circles in Figure 1b, 2a, 4a indicate?

- The heatmaps in Figure 1e, 2c, 2e, 3c, 5b are not colorblind-friendly. Could the authors change the color palette? Also, blurriness in some heatmaps makes them hard to read.

- Could the authors add a new column in the metadata of Supplementary Table 1 to distinguish between DM-DFU healers and non-healers?

- The legend "Melano/Schwan" is cut off the plot in Figure 2b.

- The analysis presented in Figure 2f is based on DiffKera or BasalKera?

- How well do the cell labels in Figure 3a correlate to those established in Figure 1b? Could the authors show an alluvial plot to check that subsetting the cells did not have major effects on how they cluster together? The same analysis could be done for the NK/NKT/T cells separate study.

- What is the difference between dashed and continuous arrows in Figure 3e?

Response to the Reviewers

We thank the reviewers for their positive opinion and constructive comments.

We have made every effort to address all raised points of concern by performing additional experiments and analyses, reworking the figures and expanding the discussion. All changes are highlighted red in the revised manuscript. Our specific point-by-point response is listed below.

Reviewer #1 (Remarks to the Author):

This manuscript by Theocharidis et al. profiled up to 200k cells from foot and forearm skin biopsies as well as PBMCs from diabetic foot ulcer patients (n=11) and non-diabetes patients (n=10) to assess the molecular and cellular associations between diabetes and ulcer healing. The manuscript consists of high-quality characterization of single-cell and spatial transcriptomic analysis of various cell types involved including macrophages and "Healing Enriched" fibroblasts. While the manuscript is largely descriptive in nature, it provides a comprehensive and informative cellular atlas of human DFU patient tissue.

Minor points:

- Would the authors be able to provide basic clinical information of all patients from whom tissues were collected, e.g., age, racial/ethnic information, gender, and other pre-existing medical conditions most notably cardiometabolic parameters, as available? This can be shown in a supplemental table.

Response: We apologize for this oversight. We have now included a new Supplementary Table 1, which contains the requested clinical information as available and have updated Supplementary Table 2 (previously Supplementary Table 1) that shows comparisons in clinical characteristics among the studied groups.

- In Figure 5G and H, the gene names in the heatmap and the Circos plot are difficult to read. Would there be a way to show a selected number of representative ligand-receptor pairs that authors would like to highlight the most in a more visible format?

Response: In order to make the figures more readable, we have adjusted the 'ligand-target regulatory potential' cutoff in the revised manuscript to reduce the number of ligand-target pairs included in the heatmap (5G) and Circos plot (5H). This resulted in reduction in the number of targets selected in the revised figure 5H, but still preserved the strongest ligand-target associations. The figures utilizing the original cutoffs have now been moved to the supplemental material as Figures 8 & 9, where they can be viewed at a more appropriate resolution.

- Have authors tried any of the available cellular trajectory and/or RNA velocity pipelines on Fibro/HE-Fibro-SMC populations to assess whether there is a transdifferentiation event among these cell types?

Response: We thank the reviewer for the suggestion. We applied the scVelo method to compute the RNA velocity of Fibro, HE-Fibro, SMC and Macrophage populations in DFUs of Healers and Non-Healers.

See new Fig. 5h for RNA velocity results and new Fig. S4 for latent time plots. The updated methods (Page 7, lines 20-35), results (Page 21, lines 13-28) and discussion (Page 24, lines 36-41) are included in the revised manuscript.

Reviewer #2 (Remarks to the Author):

The authors conducted scRNASeq in skin tissue and PBMCs from non-DM patients and DM patients with or without DFU, and found that a unique population of fibroblasts with high expression of genes related to ECM remodeling and immune/inflammation were over-represented in the samples from DFU-healers. Moreover, the healing associated fibroblasts were more like to presented in wound/ulcer bed rather than wound edge or non-wounded skin. Except for fibroblasts, authors also revealed that more M1 macrophages and naïve and central memory T-cells were presented in healers, while more M2 cells and NK and NKT cells in non-healers.

Overall, the study is well-designed, the findings were interpreted appropriately, and like the authors said that their results would be valuable sources for other related researches. But I still have one question. As the wound healing is a complex process consisting of multiple overlapping stages, is that possible the differences of cell type/gene expression pattern between DFU-healers and non-healers were the same with the differences between the last stage of healing process and the former stages? In another words, DFU-non-healers might be stuck in certain stages of wound healing rather than present as a new pattern of healing. Please discuss this question.

Response: This is a valid point. We believe that DFU-Non-healers are stagnating without progressing to the next phase of healing based on the observation that Non-healers have ongoing low-grade inflammation at the wound-site. DFU-Healers on the other hand, display an acute-like inflammatory response, which appears to be essential for successful wound repair. Indeed, the additional comparative analysis we performed using a publicly available dataset of acute wound healing demonstrated significant enrichment of HE-Fibro and M1 gene sets on Day 4-8 wounds, which correspond to the inflammation and early proliferation stages of healing and thus provided confirmatory evidence that DFU-Healers have advanced forward in the wound healing process. Therefore, it is fair to assume that what we are seeing is not a distinct new pattern of healing, but the difference between a dysregulated chronic inflammatory environment and wounds that have progressed to the first (inflammatory) or early second (proliferative) phases of healing. We would like to emphasize that one of the main significances of the study is that it provides information about the processes that are associated with DFU healing. Identifying mechanisms that can activate or guide these processes could lead to the development of new therapeutic options.

The methods (Page 7, lines 37- 46 and Page 8, lines 1-13) and results (Page 21, lines 29- 39 and Page 22, lines 1-3) have been modified to include the comparative analysis with the acute wound healing data set. Also, see new Fig. S6. The discussion has also been modified to comment on the above (Page 24, lines 36-41).

Reviewer #3 (Remarks to the Author):

The authors made a profound effort to understand the microenvironment of healing and non-healing wounds at two different locations in diabetic foot ulcers using a single cell RNA seq-approach. They created a landscape of different cells potentially interacting within the environment of healing and non-healing diabetic wounds. They found uniquely enriched fibroblast and M1 macrophage populations in diabetic foot ulcers and associated them with better healing. This study provided comprehensive correlative data but still lacks mechanistic details and conclusive evidence that truly advances our understandings of the non-healing diabetic foot ulcers. I have few major concerns on this study:

1) It is very difficult to compare healing and non-healing diabetic foot ulcers only for one time point. Impaired healing may lag behind relatively when compared to healing wounds. Therefore, a time kinetic would be important. From previous publications there is evidence that M1 macrophages play a role in removal of debris and counteract infections at the wound site. A switch to the M2 subtype of macrophages is essential for wound healing. This switch does not occur in murine diabetic wounds (Mirza R., Koh T. J. (2011). Dysregulation of monocyte/macrophage phenotype in wounds of diabetic mice and human diabetic foot ulcers. Cytokine 56, 256–264. 10.1016/j.cyto.2011.06.016; Mirza, R. E et al (2013) Blocking interleukin-1 β induces a healing-associated wound macrophage phenotype and improves healing in type 2 diabetes. Diabetes 62: 2579–2587; Lim et al. (2015) relative expression of proinflammatory and antiinflammatory genes reveals differences between healing and non healing human chronic diabetic foot ulcers. J Invest Dermatol 135:1700-1703.). This example demonstrates the importance of a time kinetic for single cell seq analysis of diabetic healing and non-healing wounds. The analysis of only one time point is prone to misinterpretation.

Response: We agree with the reviewer that analyzing longitudinal samples from the same ulcers would provide more granularity into the progress of diabetic wound healing molecular landscape. However, we are afraid that this is not feasible for the following reasons:

The DFU specimens obtained for scRNA-seq in our unit are surgically resected discarded specimens that include the whole DFU and consist of sufficient wound and periwound tissue. It should be emphasized that these are not the typical debridement specimens like the ones reported in the referenced study (Nassiri et al., J Invest Dermatol, 135:1700-1703, 2015). According to our experience, debridement samples are usually of low quality: mostly made up of callus and necrotic tissue, taken from the wound periphery and thus do not include any tissue for the wound center, where the described HE-fibroblasts were found. In pilot experiments in our unit, the Joslin-Beth Israel Deaconess Foot Center, we studied two samples from two different patients, which were collected during regular wound debridement, similar to the way samples were obtained in the above paper. Our results showed that the measured cell viabilities in the two samples were 18% and 27%, which are unacceptable for scRNA-seq. In contrast, all the single cell suspension preps analyzed from the surgically resected samples obtained for the present study had excellent viability at the time of tissue collection (typically >90%) and low levels of mitochondrial gene expression which is an established indicator of apoptotic or lysing cells. Importantly, the only published report of scRNA-seq in debrided DFU sample mentions that out of a single cell suspension of 139,000 cells, only 384 – a striking 0.27% of all isolated cells- passed QC for analysis

(Januszyk et al., *Micromachines*, 11(9): 815, 2020), which clearly demonstrates the unsuitability of debrided tissue for single cell sequencing.

In addition to the above, the highly inconsistent quality of DFU debrided tissue has been previously described (Stojadinovic et al., *Exp Dermatol*, 22(3): 216–218, 2013), with the authors showing that only half of examined samples had acceptable RNA integrity numbers (RIN), which is a measure of RNA quality, for further microarray or next generation sequencing analyses. This substantial attrition rate would require enrolment of extra patients, still without ensuring that the multiple time point debridement samples meet the criteria for downstream analysis.

Furthermore, in acute wounds the healing process is linear and tightly regulated and can be experimentally studied in human subjects by using specific sizes of punch biopsies to inflict a wound at the exact same anatomical site and subsequently collect the wounded tissue at defined time points. Conversely, DFUs are inherently highly variable, with different sizes, different anatomical locations on the foot and different starting points, derived from a patient population with multiple divergent comorbidities. All these factors render a time course analysis of diabetic wound healing especially problematic and samples collected at the same time points of the study (e.g. week 2, 3 etc.) wouldn't necessarily correspond to the same point in healing, even if belonging to the same clinical group.

It can be claimed that in order to avoid all pitfalls related to the analysis of debrided tissue during regular office visits by the patients, the only alternative would be to take wound biopsies sizable enough to allow sc-RNAseq analysis. However, this approach carries significant risks to the patients as it can impede wound healing and even lead to severe complications, such as amputations. This results in highly unfavorable risk/benefit ratio making such studies unethical and therefore cannot be approved by the IRB.

Support to the above can be seen by the fact that the vast majority of studies looking into the gene expression of DFUs include a single time point: Li et al., *J Invest Dermatol*, 137(12):2630-2638, (2017) analyzed 29 DFU patients with qRT-PCR; Ramirez et al., *J Invest Dermatol*, 138(5): 1187–1196, (2018) analyzed 14 DFU patients with microarray; Narayanan et al., *Commun Biol*, 3(1):768, (2020) analyzed 8 DFU patients with qRT-PCR; Sawaya et al., *Nat Commun*, 11(1):4678, (2020) analyzed 13 DFU patients with RNA-seq.

We have modified the discussion to comment on this Page 25, lines 1-7.

2) Authors should analyse foot healing over a time period in particular enrichment kinetics of different cells population and associated gene signatures. Analysis of wound biopsies from diabetic and acute wounds collected at different time points would solidify their conclusions. It would be interesting to see whether diabetic healing wound closely overlap with healthy acute human wounds in terms of fibroblasts, keratinocytes, T cells and macrophages recruitment and transcriptome. It is also not clear what drives accumulation of uniquely enriched fibroblasts or M1 macrophages in healing foot ulcers and whether healing fibroblasts or immune cells influence each others recruitment in a paracrine manner.

Response: As mentioned above, the only way to analyze a diabetic foot ulcer over various time points is to take large enough wound biopsies, as analyzing discarded debrided specimens is not adequate. However, the risk to the patients, including delay or failure to heal their ulcer, is unacceptable and

cannot get IRB approval. This is the main reason that previous studies have only employed discarded debrided tissue specimens that are mainly comprised of callus and necrotic tissue and are not suitable for this type of analysis. Furthermore, even if biopsies were feasible, they would only involve the wound edges and in all probability would miss the wound bed area where the healing enriched fibroblasts were observed.

The primary objective of our study was not to construct an atlas of the healing time course in DFUs, but rather identify potential gene targets and cell populations that differentiate healing vs non-healing ulcers. Diabetic wound healing in humans is such a highly dysregulated and irregular process and our reasoning for not including multiple time points is both technical and logical as we explained in the previous point response.

Performing new scRNA-seq analysis on human non-diabetic acute wounds, even though highly informative, is beyond the scope of this manuscript and will be considered for future studies. There is no such dataset publicly available, but we have used published longitudinal microarray data to perform comparative analysis with our findings.

We agree that it will be important to define what drives the accumulation of HE-Fibro as well as potential interactions with other cell types and future experiments will examine paracrine interplay of macrophages and HE-Fibro employing co-culture *in vitro* systems or conditioned media treatment.

We thank the reviewer for the suggestion to perform comparative analysis with acute wound healing data and in the revised manuscript we have harnessed a previously published longitudinal dataset to explore the transcriptomic signature of healing associated cell types from our observations over time in acute wounds.

Please also see response to Reviewer #2. This has been addressed in the manuscript methods (Page 7, lines 37-46 and Page 8, lines 1-13), results (Page 21, lines 29-39 and Page 22, lines 1-3), discussion (Page 24, lines 36-41) and new Fig. S6.

3) Authors should characterize M1 and M2 macrophages in more detail and experimentally explain how accumulated M2 macrophages oppose healing of diabetic foot. What is the origin or source of these distinct cells type whether they proliferate or differentiate locally or were recruited from distant organs. Some aspects of this manuscript are not entirely novel, as some of the transcriptional findings of human diabetic foot ulcers and dysregulated inflammation has already been reported by Sawaya et. al. (Nat Commun . 2020 Sep 16;11(1):4678.).

Response:

Skin wound macrophages are either tissue resident and differentiated from bone marrow-derived monocytes with a minimal contribution of yolk sac originating macrophages, or peripherally recruited monocytes responding to inflammatory cues and similarly differentiating to macrophages at the injury site. Most of the research performed on skin macrophages has utilized parabiosis chimeras, conditional and targeted cell ablation and fate-mapping models in mice and has not been translated to humans yet. Substantial studies with flow cytometry to establish potentially evolutionarily conserved sets of cell surface markers capable of differentiating among the different macrophage subsets are required in

human patients. Given that the majority of skin macrophages has a common origin of bone marrow-derived monocytes adds further complexity to the characterization attempts.

The manuscript discussion has been updated to incorporate discussion on M2 macrophages polarization and its association with wound healing as well as skin macrophage origin **Page 24, lines 12-23.**

We agree that parts of our study are not entirely novel and for this reason we have mainly focused on novel and unreported findings. We have already referenced the study by Sawaya et al. and their results are congruent with our findings.

4) Authors should provide more data on how activated IL-6 signalling drives healing of diabetic wound, as this previously have been reported to promote local inflammation and prevent wound healing.

Response: We have elaborated on the importance of IL6 in diabetic wound healing in the revised manuscript **Page 20, lines 1-5.**

5) Validation of selected biological pathways shown in Figure 4 and Figure 5 in diabetic foot ulcers as well as diabetic and healthy age/gender matched foot pad skin is required at protein level to support findings at transcriptional levels. Author are recommended to causally validate some of the key findings in diabetic mice wound healing model using genetic or pharmacological approaches.

Response: We agree that it's important to confirm gene expression data at the protein level and in the revised manuscript we have performed new experiments to validate selected top activated pathways and regulators with immunostaining. The methods have been updated (**Page 8, lines 32-46 and Page 9, lines 1-15**). In new Fig. S3 we present immunofluorescent staining and quantification for macrophage markers and for activated regulators from Figs. 4 and 5. Since our priority and most important comparisons were between DFU-Healers and Non-healers we focused our analysis on these specific groups. The results have been updated to describe findings from new experiments (**Page 18, lines 1-5; Page 18, lines 10-11; Page 20, lines 10-13**).

We would also like to point out that although *in vivo* mouse models are important, they are beyond the scope of the present study and will be performed in future experiments.

6) More detailed protein analysis including chemokine, cytokines and their major regulators would help to better understand the wound microenvironment. Multiplex immunostaining of different fibroblasts and immune cells in conjunction with highlighted pathways proteins at different time points is required.

Response: We appreciate the reviewer's suggestion and we agree that multiplex immunostaining and in depth investigation of chemokines, cytokines and regulators would better characterize the wound microenvironment. Nevertheless, it is unrealistic to perform all these additional extensive experiments for the present manuscript. We are confident that the quality and quantity of the presented data is sufficient to justify publication while, at the same point, as every other paper, raises a number of questions that will need to be explored in future studies.

7) Authors should provide demographic details of the patient samples. (Duration of diabetes mellitus, wound size, HbA1c, comorbidities, age, medication). This would be best shown in a table.

Response: We have now included Supplementary Table 1 with all this information.

Reviewer #4 (Remarks to the Author):

In this paper, Theocharidis et al investigate the diabetic wound healing microenvironment by comparing the transcriptomic profile of the PBMCs, foot and forearm in diabetic patients suffering from diabetic foot ulcers (DFU) that heal (healers), diabetic patients suffering from DFU that cannot heal (non-healers), diabetic patients with no DFU (DM patients), and non-diabetic patients (non-DM patients). The authors perform scRNA-seq analysis at different levels, first with all the cells sequenced together and later performing the analysis to PBMC samples only, to NK/NKT/T cells only, to cells from the foot only, and to fibroblasts only. With this approach, a fibroblast cell type associated with wound healing is reported, and its existence and localization is validated with spatial transcriptomics. In addition, differences in the population ratios of immune cells are also detected between the different clinical groups.

The paper is well written and easy to follow. The findings are well justified and will be useful for investigators (and hopefully patients!) in the diabetic research field.

However, the data is presented in a very qualitative approach and it is hard to trust some of the reported enrichments in terms of cell type population fractions. I hope that by addressing the major comments below, the paper becomes more quantitative.

Major comments:

- In the methods, the authors state that the significance testing change in abundance of cell types across clinical groups was performed either using ANOVA or Welch's t-test. However, no p-values are reported in several barplots (e.g. Figure 1d), leaving it to the reader to assess/believe significant enrichments of cell types in the different clinical groups. It would be good to indicate enrichment p-values in the figures.

Response: In the revised manuscript we have included new Fig. S1 showing the comparative analysis of cellular abundance for each cell type across clinical groups from all samples across all three anatomical sites. The results have been updated (Page 13, lines 26-28). We have also modified Figure 1d to display statistical significance. In addition, we adapted Figure 3d and Figure 4b, c, d to provide more clarity with regard to statistical comparisons and significances in the specific anatomical sites of interest.

- In Figure 1d, erythros and plasma seem to be enriched in non-healers, but nothing is stated. Additionally, Mast cells also seem enriched in the controls. Is there any biological or technical reason for it?

Response: We thank the reviewer for pointing out this omission. This point has been addressed at Page 13, lines 28-34.

- In the first section of the results, what do the authors mean by "orthogonal clusters"?

Response: The term orthogonal means that the clusters are statistically independent.

- Being a clinical study, I expect there to be batch effects because of patient of origin. Could the authors show whether HE-Fibros are equally present in all healer patients?

Response: Figure 4b has been modified to include points showing HE-Fibro counts for each patient. All DFU-Healers expect for 1 patient has significantly higher number of HE-Fibro as compare to Non-healers as well as skin from healthy controls and diabetic patients.

- How are the error bars in Figure 3b computed?

Response: In Figure 3b, the error bars represent standard error of mean. The figure legend has been amended to include this information.

- What is the rationale to select cluster 3 as the “sender” population in the NicheNetR analysis perform in Figure 5?

Response: Out of the four HE-Fibro clusters (3, 4, 6, 13), cluster 3 had the highest expression of inflammatory and remodeling cytokines (IL6, CHI3L1, MMP1). Given the importance of these class of signaling molecules in healing, we opted to choose cluster 3 as the primary ‘sender cluster’ to generate a list of ligand candidates for the NicheNetR analysis. This is clarified in the text Page 20, lines 24-27.

- The authors perform scRNA-seq analysis on multiple samples from the same patient. Which clinical group does the patient belong to (I assume it is a healer)? Has this data been produced 3 weeks after previous scRNA-seq data? Can the authors extract any information about the population dynamics of the HE-fibro cell type (e.g. does their fraction increase/decrease during the course of wound healing?)

Response: As mentioned in the text Page 22, lines 39-40 and Page 23, line 1: “ScRNA-Seq analysis was performed on skin specimens of the same patient from three different sites: wound bed, wound edge, and non-wound excess skin from a pressure sore excision.”

All the samples were thus collected at the same time and the patient had a different type of chronic wound, hence they do not belong to any of the clinical groups. This was a confirmatory approach to show localization of HE-Fibro within the wound bed and the conclusion we reached as mentioned in the text Page 23, lines 7-9:

“These results further affirm an association of HE-Fibro with the wound healing process in an additional type of chronic wound and point toward heterogeneity of fibroblasts across different regions of ulcers.”

Minor comments:

- In the abstract, the authors use the acronyms PBMC and DFU without previous definition. This can be confusing for non-experts.

Response: We apologize for the oversight; this has now been corrected in the revised manuscript abstract.

- Did the authors consider getting skin biopsies from the foot 12 weeks post-surgery in healers and non-healers to investigate dynamic changes in the wound microenvironment? Could this reveal some temporal differences in the establishment of fibroblast populations required for wound healing?

Response: This is a great suggestion. However, getting skin biopsies of high enough quality, would essentially mean re-wounding the patients, which is non-advisable for prone to chronic wound development populations. Please also see response to Reviewer #3 points 1 and 2.

- The website (<https://bhasinlab.bmi.emory.edu/Diacomp/>) is not user-friendly: tick-boxes overlap with the UMAP's legend unless the user displays the project summary. When the project summary is displayed, tick-boxes are too far from the UMAP and the user needs to keep scrolling up and down. Can this be optimized? Additionally, can Supplementary Table 2 be made accessible via the website?

Response: The new Supplementary table 4 (old Supplementary Table 2), "Distribution of different cell types across anatomical locations", has now been included in the online tool, which has also been re-worked to improve usability. When the Diacomp viewer is accessed on a smaller screen, the tick-boxes and UMAP may overlap due to limited screen space. If the user zooms out on their browser page, they will be able to access the tool more easily.

- What do the dashed circles in Figure 1b, 2a, 4a indicate?

Response: The dashed circles in Figures 1b, 2a, and 4a indicate similar cell groups of a particular lineage.

- The heatmaps in Figure 1e, 2c, 2e, 3c, 5b are not colorblind-friendly. Could the authors change the color palette? Also, blurriness in some heatmaps makes them hard to read.

Response: Thank you for pointing this out. We have updated the heatmaps with a colorblind-friendly palette and increased the resolution.

- Could the authors add a new column in the metadata of Supplementary Table 1 to distinguish between DM-DFU healers and non-healers?

Response: In the revised supplementary Table 2, we have added this information.

- The legend "Melano/Schwan" is cut off the plot in Figure 2b.

Response: Thanks, we corrected this in the revised Figure.

- The analysis presented in Figure 2f is based on DiffKera or BasalKera?

Response: The analysis in Figure 2f includes both DiffKera and BasalKera groups.

- How well do the cell labels in Figure 3a correlate to those established in Figure 1b? Could the authors show an alluvial plot to check that subsetting the cells did not have major effects on how they cluster together? The same analysis could be done for the NK/NKT/T cells separate study.

Response: In Figure 1, we labeled major cell types without subgrouping. In Figure 3a, to increase resolution, we labeled specific subtypes of cells. Across the paper, to label the major cell types, we used the same immune cell markers that are in Figure 1c.

- What is the difference between dashed and continuous arrows in Figure 3e?

Response: In Figure 3e, the dashed arrows represent an indirect interaction, and the continuous arrows represent a direct physical molecular interaction. In the revised manuscript, we have updated Figure 3e to include a legend clarifying all the different IPA annotations.

Reviewers' Comments:

Reviewer #1:

None

Reviewer #2:

Remarks to the Author:

The authors revised the manuscript appropriately. No further comments.

Reviewer #3:

Remarks to the Author:

The authors have addressed all of my comments and criticisms in a satisfying manner.

Reviewer #4:

Remarks to the Author:

In their rebuttal, Theocharidis et al address all my concerns. In particular, the significance testing change in abundance of cell types across clinical groups becomes more clear in the new figures. I don't have any other comment.

Response to the Reviewers

REVIEWERS' COMMENTS

Reviewer #2 (Remarks to the Author):

The authors revised the manuscript appropriately. No further comments.

We thank the reviewer for their positive comments.

Reviewer #3 (Remarks to the Author):

The authors have addressed all of my comments and criticisms in a satisfying manner.

We thank the reviewer for their positive comments.

Reviewer #4 (Remarks to the Author):

In their rebuttal, Theocharidis et al address all my concerns. In particular, the significance testing change in abundance of cell types across clinical groups becomes more clear in the new figures. I don't have any other comment.

We thank the reviewer for their positive comments.